# Host cystathionine-γ lyase derived hydrogen sulfide protects against *Pseudomonas aeruginosa* sepsis

Georgios Renieris[1], Dionysia-Eirini Droggiti[1], Konstantina Katrini[1],
Panagiotis Koufargyris[1], Theologia Gkavogianni[1], Eleni Karakike[1],
Nikolaos Antonakos[1], Georgia Damoraki[1], Athanasios Karageorgos[1],
Labros Sabracos[1], Antonia Katsouda[2], Elisa Jentho[3,4,5], Sebastian Weis[3,4,5], Rui Wang[6],
Michael Bauer[3], Csaba Szabo[7,8], Kalliopi Platoni[9], Vasilios Kouloulias[9],
Andreas Papapetropoulos[2,10], Evangelos J. Giamarellos-Bourboulis[1]*

1 4th Department of Internal Medicine, National and Kapodistrian University of Athens, Medical School, Athens, Greece, 2 Center of Clinical, Experimental Surgery & Translational Research, Biomedical Research Foundation of the Academy of Athens, Athens, Greece, 3 Department of Anesthesiology and Intensive Care, Jena University Hospital, Jena, Germany, 4 Institute for Infectious Disease and Infection Control, Jena University Hospital, Jena, Germany, 5 Center for Sepsis Control and Care, Jena University Hospital, Jena, Germany, 6 Department of Biology, York University, Toronto, Canada, 7 Department of Anaesthesiology, University of Texas Medical Branch, Galveston, Texas, United States of America, 8 Chair of Pharmacology, Department of Medicine, University of Fribourg, Fribourg, Switzerland, 9 2nd Department of Radiology, National and Kapodistrian University of Athens, Medical School, Athens, Greece, 10 Laboratory of Pharmacology, Faculty of Pharmacy, National and Kapodistrian University of Athens, Medical School, Athens, Greece

* egiamarel@med.uoa.gr

**Data Availability Statement:** All relevant data are within the manuscript and its Supporting Information files.

## Abstract

Hydrogen sulfide ($H_2S$) has recently been recognized as a novel gaseous transmitter with several anti-inflammatory properties. The role of host-derived $H_2S$ in infections by *Pseudomonas aeruginosa* was investigated in clinical and mouse models. $H_2S$ concentrations and survival was assessed in septic patients with lung infection. Animal experiments using a model of severe systemic multidrug-resistant *P. aeruginosa* infection were performed using mice with a constitutive knock-out of cystathionine-γ lyase (*Cse*) gene (*Cse*-/-) and wild-type mice with a physiological expression (*Cse*+/+). Experiments were repeated in mice after a) treatment with cyclophosphamide; b) bone marrow transplantation (BMT) from a *Cse*+/+ donor; c) treatment with $H_2S$ synthesis inhibitor aminooxyacetic acid (AAA) or propargylglycine (PAG) and d) $H_2S$ donor sodium thiosulfate (STS) or GYY3147. Bacterial loads and myeloperoxidase activity were measured in tissue samples. The expression of quorum sensing genes (QS) was determined in vivo and in vitro. Cytokine concentration was measured in serum and incubated splenocytes. Patients survivors at day 28 had significantly higher serum $H_2S$ compared to non-survivors. A cut-off point of 5.3 μM discriminated survivors with sensitivity 92.3%. Mortality after 28 days was 30.9% and 93.7% in patients with $H_2S$ higher and less than 5.3 μM (p = $7 \times 10^{-6}$). In mice expression of *Cse* and application of STS afforded protection against infection with multidrug-resistant *P. aeruginosa*. Cyclophosphamide pretreatment eliminated the survival benefit of *Cse*+/+ mice, whereas BMT increased the survival of *Cse*-/- mice. *Cse*-/- mice had increased pathogen loads compared

**Funding:** EJG-B has received honoraria (paid to the University of Athens) from AbbVie USA, Abbott CH, Angelini Italy, Biotest Germany, InflaRx GmbH, MSD Greece and XBiotech Inc. He has received independent educational grants from AbbVie, Abbott, Astellas Pharma, AxisShield, bioMérieux Inc, InflaRx GmbH and XBiotech Inc. He has received funding from the FrameWork 7 program HemoSpec and from the Horizon2020 Marie-Curie project European Sepsis Academy (granted to the National and Kapodistrian University of Athens). GR and EK are funded by the European Union's Horizon 2020 Marie Skłodowska-Curie grant European Sepsis Academy No 676129. CSz is supported by SNSF (Swiss National Research Foundation). MB is funded by the Deutsche Forschungsgemeinschaft (DFG, German Research Foundation) under Germany's Excellence Strategy – EXC 2051 – Project-ID 390713860. The funders had no role in study design, data collection and analysis, decision to publish, or preparation of the manuscript.

**Competing interests:** The authors have declared that no competing interests exist.

to $Cse^{+/+}$ mice. Phagocytic activity of leukocytes from $Cse^{-/-}$ mice was reduced but was restored after H$_2$S supplementation. An H$_2$S dependent down- regulation of quorum sensing genes of *P.aeruginosa* could be demonstrated *in vivo* and *in vitro*. Endogenous H$_2$S is a potential independent parameter correlating with the outcome of *P. aeruginosa*. H$_2$S provides resistance to infection by MDR bacterial pathogens.

## Author summary

Multidrug- resistant *Pseudomonas (P.) aeruginosa* is one of the most common causes of ventilator-associated pneumonia (VAP), which consist one of the most common causes of sepsis in hospitalized patients. The increasing antimicrobial resistance of *P. aeruginosa* necessitates the investigation of new treatment strategies. In this study we highlight host derived hydrogen sulfide (H$_2$S), produced by cystathionine-γ-lyase (CSE), as a new protective mechanism against *Pseudomonas aeruginosa* infection. We have shown that serum H$_2$S levels above 5.3μM can discriminate survivors of *P.aeruginosa* related sepsis with high sensitivity and is correlated with reduced mortality. We verified this finding in an animal model of CSE knockout mice. The protective role of endogenous H$_2$S resides in the enhancement of neutrophil recruitment and the phagocytic activity of neutrophils, as well as by inhibiting the cell-to-cell communication system of *P. aeruginosa* (quorum sensing system), rendering it more vulnerable to phagocytosis.

## Introduction

Ventilator-associated pneumonia (VAP) is a common and life-threatening complication of mechanical ventilation and one of the most common causes of sepsis in hospitalized patients. *Klebsiella (K.) pneumoniae*, *Acinetobacter (A.) baumannii* and *Pseudomonas (P.) aeruginosa* are responsible for more than 50% of VAP cases in Greece and are characterized by high rates of antimicrobial resistance [1]. This mandates the investigation of new strategies, which can tackle the problem of antimicrobial resistance and lead to resolution of VAP.

Hydrogen sulfide (H$_2$S) is the third gaseous transmitter -after nitric oxide and carbon monoxide- that is produced mainly through the action of three distinct enzymes cystathionine γ-lyase (CSE), cystathionine β- synthase (CBS) and 3- mercaptopyruvate sulfurtransferase (3MST) [2]. In mammals, *Cse* is constitutively expressed in many peripheral organs and is upregulated in response to cellular stress. CSE-derived H$_2$S promotes angiogenesis, drives vasorelaxation, reduces atherosclerosis and prevents ischemia-reperfusion injury [2,3]. Genetic ablation or pharmacological inhibition of CSE promotes leukocyte adherence to the vessel wall and increases cytokine production [4,5]. The pro-inflammatory effects observed after inhibition of CSE led us study the levels of host H$_2$S in cases of lower respiratory tract infections (LRTIs) including ventilator associated pneumonia (VAP) and pneumonia caused by the novel SARS-CoV-2 virus (COVID-19 disease). Results showed that plasma H$_2$S levels are higher among survivors of LRTI by COVID-19; non-survivors were characterized by disability to increase plasma H$_2$S over the first seven days of follow-up [6].

In this study we focused on the selective role of host H$_2$S in the pathogenesis of infections by *P. aeruginosa*. Using a two-step approach, we demonstrate that host-H$_2$S is associated with the outcome of VAP caused by *P. aeruginosa* but not by other pathogens. We then investigated the underlying mechanism of action of host derived H$_2$S in an animal model of infection by MDR *Pseudomonas aeruginosa*.

## Materials and methods

### Ethics statement

The Hellenic Sepsis Study Group (HSSG) is collecting clinical data and biosamples from patients with sepsis since 2006 from 65 study sites across Greece (intensive care units, emergency departments and departments of internal medicine or surgery; www.sepsis.gr). The patients are enrolled after written consent provided by themselves or by their first-degree relatives (for patients unable to consent). The protocol is approved by the Ethics Committees of the participating hospitals.

Animal experiments were conducted in the unit of animals for medical scientific purposes of ATTN University General Hospital (Athens, Greece) according to EU Directive 2010/63/EU and to the Greek law 2015/2001, which incorporates the Convention for the Protection of Vertebrate Animals used for Experimental and Other Scientific Purposes of the Council of Europe (code of the facility EL 25BIO014, approval no. 1853/2015). All experiments were licensed from the Greek veterinary directorate under the protocol number 5591/19-10-2017.

### Clinical study

Adults with clinically or microbiologically documented infections and at least two signs of the systemic inflammatory response syndrome (SIRS) are enrolled. Since March 2016 all patients in the database were re-classified into non-sepsis, sepsis and septic shock using the Sepsis-3 classification criteria [7]. Exclusion criteria were: a) HIV-1 infection; b) neutropenia defined as less than 1,000 neutrophils/mm$^3$; and c) chronic use of corticosteroids defined as more than 1 mg/kg/day of prednisone equivalent for more than 15 days. Infections and organ dysfunction are defined according to already-published international criteria [8]. Blood sampling is done within the first 24 hours from enrollment.

The following variables are recorded: i) demographics; ii) Acute Physiology and Chronic Health Evaluation (APACHE) II score, Charlson Comorbidity Index (CCI) and Sequential Organ Failure Assessment (SOFA) score; (iii) biochemistry, absolute blood cell counts and arterial gases; iv) appropriateness of the administered antimicrobial treatment defined as the susceptibility of the isolated pathogen to at least one of the empirically prescribed antimicrobials according to the antibiotic susceptibility testing; (v) quantitative cultures of blood, urine and tracheobronchial secretions (TBS) performed on baseline day 1.

Among patients entered in the database, patients with microbiologically confirmed ventilator associated pneumonia (VAP) due to monomicrobial infection by *P. aeruginosa* (group A); *K. pneumoniae* (group B) and *A. baumannii* (group C), were selected. Pathogens grew at $\geq 10^5$ cfu/ml in tracheobronchial secretions. VAP was defined according to already-published international criteria [8]. Survival was recorded for 28 days. 10 ml of blood was collected after peripheral venipuncture within the first 24 h of enrollment. Serum was prepared by centrifugation at 900 g, and samples were transported within the same day to the central lab and stored at– 80˚C until processing. Blood samples were also isolated from healthy volunteers, who were matched with patients with VAP for age and gender.

### H$_2$S measurements

H$_2$S was measured in the serum samples of healthy volunteers and patients by using monobromobimane (MBB) derivation followed by high performance liquid chromatography as previously described [6,9,10]. MBB, monosodium phosphate (NaH$_2$PO$_4$), disodium phosphate (Na$_2$HPO$_4$) and 5-sulfosalicylic acid (SSA) were purchased from Sigma-Aldrich (St. Lewis MO, USA). Sodium sulfide (Na$_2$S), diethylenetriaminepentaacetic acid (DTPA) and

trifluoroacetic acid (TFA) were purchased from Alfa Aesar (Erlenbachweg, Germany). Tris-HCl buffer (0.1 M pH 9.5) was purchased from AlterChem (Athens, Greece). HPLC grade acetonitrile (ACN) and water were purchased from Scharlab (Sentmenat, Barcelona, Spain). Phosphate buffer pH 8.5 20 mM was prepared by dissolving $NaH_2PO_4$ and $Na_2HPO_4$ in distilled water. For the preparation of the derivatization buffer (Tris-HCl 0.1 M pH 9.5, 0.1 mM DTPA) DTPA was dissolved in tris-HCl. All solvents and buffers, as well as the tubes used for the derivatization reaction, were deoxygenated by using nitrogen gas flow (10mins and 30secs respectively). The solutions for the sulfide standard curve were prepared by dissolving $Na_2S$ in phosphate buffer, to final concentrations of 4–250 μM and the quantification limit was 1 μM. The MBB 10 mM derivatization solution was prepared by dissolving MBB in ACN, was then aliquoted in dark containers and kept at -20° C. The SSA 200 mM stop solution was freshly prepared before each measurement by dissolving SSA in distilled water. All sample preparations were conducted under dim room lighting. After deoxygenation, 30 μl of serum sample or standard, 70 μl Tris-HCl 0.1 M pH 9,5 0.1 mM DTPA and 50 μl MBB 10 mM were added in the tubes. The mixture was incubated at hypoxic conditions (1% $O_2$) at 37° C for 60 sec. The derivatization reaction was stopped by adding 50 μl of 200 mM SSA, followed by vortexing for 10 secs. The vials were then left on ice for 10min and centrifuged at 12000 rpm, 4° C for another 10 mins. Finally, 100 μl of the supernatant were transferred at darkened HPLC vials and kept at 4˚C. The chromatographic experiments were conducted on an Agilent 1100 HPLC system (Agilent, Waldbronn, Germany). The separation was performed on a LiChroCART Reverse-Phase (RP) C18 4.6 x 250 mm, 5 μm analytical column, with a Purospher RP-18E 4 x 4 mm, 5μM guard column, both obtained from Merck (Darmstadt, Germany). Analysis was performed at 25˚C, using gradient elution (Table B in S1 Text). The two mobile phases consisted of ACN (A, 0.1% TFA, v/v) and water (B, 0.1% TFA), v/v), at a 0.6 ml/min flow rate. The sample injection volume was 20 μl. All solvents were filtered through a 0.45 μm membrane filter (Agilent). All measurements were carried out at excitation and emission wavelengths of 390 nm and 475 nm respectively. The retention time of the derivatization product was 12.7 minutes.

## Animal studies

**Pathogens.** Three different *P. aeruginosa* isolates from tracheobronchial secretions of patients with sepsis due to VAP were used; a) the *P. aeruginosa* 6–11–19 isolate with minimal inhibitory concentration (MIC) of amikacin, ceftazidime, ciprofloxacin and meropenem >256, >512, >16 and 8 μg/ml respectively; b) the *P. aeruginosa* 19–2–45 isolate 2 with MIC of amikacin, ceftazidime, ciprofloxacin and meropenem >256, >512, >128 and >256 μg/ml respectively; and c) the *P. aeruginosa* 20–1–30 isolate with MIC of amikacin, ceftazidime, ciprofloxacin and meropenem >256, >512, >64 and >64 μg/ml respectively. solates were genetically distinct as defined by pulsed-field gel electrophoresis (PFGE) of their DNA to exclude similar isolates coming from horizontal spread (S3A Fig). Additionally, a *K. pneumoniae* 87B producing carbapenemase KPC-2 isolate with MIC of amikacin, meropenem, aztreonam and colistin >512, >512, 128 and 32 μg/ml respectively. The MIC was measured with the microdilution method and the production of KPC-2 by polymerase chain reaction.

**Animals.** We used 406 male and female mice (7–8 weeks old). Mice were allowed to acclimate for seven days before beginning the experiments. They were housed in individually ventilated cages, up to 5 mice per cage on 12-h dark/light cycle and allowed free access to standard dry rodent diet and water, supplemented with 1g/l L- cysteine (AppliChem, Darmstadt, Germany) [11]. Analgesia was achieved with paracetamol suppositories, in order to avoid interactions with the immune system.

H$_2$S synthesis is catalyzed by three different enzymatic systems, cystathionine γ- lyase (CSE), which is the key enzyme for H$_2$S synthesis in the liver and kidney, cystathionine β-synthase (CBS), which is overexpressed in brain tissue and 3-mercaptopyruvate sulfurtransferase (3MST) which is ubiquitously expressed [12]. We investigated the role only of CSE and 3MST, due to the fact that homozygotic CBS deficiency is related to a neonatal mortality of over 90%. [13,14]. *Cse* [-/-] mice were created after breeding of *Cse* [+/-] mice (Friedrich Shiller University Hospital, Jena, Germany). The *Cse*[+/-] mice were generated as previously described [13]. In short chimeric male *Cse* [+/-] mice were produced after injection of *Cth*[+/-] stem cells into C57BL/6 blastocysts. These were then crossed with C57BL/6 female mice to obtain *Cse* [+/-] pups. *Cse* [+/-] pups were backcrossed multiple times to achieve high genetic homogeneity on C57BL/6 background. The *Cse* [+/-] males and females produced were finally bred to obtain wild- type (*Cse* [+/+]), *Cse* [+/-], and *Cse* [-/-] littermates. 3*Mst*[+/+] and 3*Mst*[-/-] mice were created, as described previously [15].

**Genotyping.**    4 weeks after birth genomic DNA for genotyping mice was prepared from the tail tips by the rapid, hot sodium hydroxide and Tris (HotSHOT) [16]. A small tail punch (<2mm) was prepared using a sharp sterile surgical scissor and the tail biopsy was placed immediately into sterile polypropylene microfuge tube and stored at -80˚C until DNA isolation and purification. The CSE null and wild-type alleles were detected by a three-primer PCR mild modified protocol [17] of tail-tip DNA: N1 (5′-TGC GAG GCC AGA GGC CAG TTG TGT AGC-3′), F1 (5′- TGT TCA TGG TAG GTT TGG CC-3′) and R1 (5′-TCA GAA CTC GCA GGG TAG AA -3′). These primers amplify a ~500 bp wild type band (from primers F1 and R1) and a ~450 bp targeted (null) band (from primers N1 and R1). All primers were purchased from IDT (IDT, San Jose, California, USA). The samples were spun down and 2 μl of the supernatant was used as a template in a 25-μl reaction using 12 μl BioMix (Bioline GmbH, Luckenwalde, Germany), 3.33 pmol/ml of each primer and 1.5–2.0 mM MgCl2. qRT-PCR was performed by the iQ5 cycler system (BioRad, Hercules, CA, USA). The thermal cycling conditions consisted of a 7 min initial denaturation cycle at 95˚C; followed by 30 cycles of denaturation at 60˚C for 75 s; annealing at 72˚C for 2 min; extension at 95˚C for 1 min. Finally, the reaction ended with 1 cycle at 60˚C for 75 s and with a final extension at 72˚C for 10 min. After this, samples were kept at 4˚C. Reactions containing Molecular Grade Water (Appli-Chem, GmbH, Germany) were used as negative controls to evaluate for potential contamination. Specificity was confirmed by electrophoretic analysis of the reaction products by 2% w/v agarose electrophoresis gel in 0.5 × Tris–Borate EDTA (TBE) buffer (5V/cm) and 1 μg/mL ethidium bromide staining (AppliChem).

**Animal studies.**    *Cse* [+/+] and *Cse* [-/-] mice were infected by intraperitoneally injecting 1x10$^8$ cfu/mouse of MDR *P. aeruginosa* isolate 6–11–19, concentration in which *P. aeruginosa* is characterized by a lethality of up to 60% in infection models in rodents [18]. Mice were then randomized via a typical randomization table in treatment groups receiving 200 μl subcutaneously of: a) 120 mg/kg of the H$_2$S biosynthesis inhibitor aminooxyacetic acid AOAA (Sigma-Aldrich), which inhibits both CBS and CSE catalytic activity [19] diluted in 0.9% NaCl once per day for 4 days or 0.9% NaCl once daily for 4 days; b) 200 μmol/kg/mouse PAG (Sigma-Aldrich), which is a specific CSE inhibitor [20], diluted in 0,9% NaCl administrated intraperitoneally once per day for 4 days; c) 2 g/kg/mouse [21] sodium thiosulfate 10% (STS, Dr. Franz Köhler Chemie GmbH, Bensheim, Germany) diluted in WFI once per day for 4 days or WFI once per day for 4 days and d) 25mg/kg/mouse [22] GYY3147 (Sigma-Aldrich). Some experiments were repeated in *CSE*[+/+] and *CSE*[-/-] mice after induction of neutropenia through i.p. injection of 150mg/kg and 100mg/kg of cyclophosphamide (Baxter, Chicago, Illinois, USA) 4 and 1 days before the MDR *P. aeruginosa* injection. Additional survival experiments were done using 1 x 10$^8$ cfu/mouse of the MDR *P. aeruginosa* isolates 19–2–45 and 20–1–30 and 1 x

$10^8$ cfu/mouse of the carbapenemase-producing *Klebsiella pneumoniae* isolate *(Lee et al., 2015)* as well as challenge with 10 or 30 mg/kg/mouse bacterial endotoxin (LPS) of *E. coli* O55:B5 (Sigma-Aldrich). Some experiments were repeated in *3Mst*$^{+/+}$ and *3Mst*$^{-/-}$ mice. On each day of experimentation, we infected up to 3 mice from each group. Survival was recorded every 12 hours for 7 days. Analgesia was achieved by subcutaneous administration of meloxicam 5 mg/kg.

*Cse*$^{+/+}$ and *Cse*$^{/-}$ mice were sacrificed at certain timepoints after infection with MDR *P. aeruginosa* isolate 6–11–19 by s.c. injection of 300 mg/kg ketamine, followed by cervical dislocation. Under sterile conditions a midline abdominal incision was performed, intestines were displaced to the left and the lower vena cava was punctured by a 20-gauge needle. 500 μl of whole blood was aspirated, collected into sterile and pyrogen- free tubes (Vacutainer, Cockeysville, MD, USA) and centrifuged. Serum was stored at -80˚C. Then, segments of the liver and of the lower lobe of the right lung were excised and collected into sterile tubes with 1 ml NaCl 0.9%. The samples were weighted and homogenized. Additionally, segments of the spleen were stored in RPMI 1640 (Biochrom, Berlin, Germany) and of the lower lobe of the right lung in RNAlater (Qiagen, Hilden, Germany).

**Bone marrow transplantation (BMT).** Survival experiments were repeated in *Cse*$^{-/-}$ and *Cse*$^{+/+}$ mice after bone marrow transplantation [23]. Recipient *Cse*$^{-/-}$ and *Cse*$^{+/+}$ mice were placed in an acrylic container (22 cm diameter, 12 cm depth) and irradiated with a single dose of 9.5 Gy at a dose rate pf 300cGy/min (Energy 6MV photons) in a VitalBeam irradiator (Varian, CA, USA). One day after the irradiation BMT was done using bone marrow from *Cse*$^{+/+}$ mice. More precisely, mice were killed and both hind legs were removed under aseptic conditions. Bone marrow cells (BMCs) were collected by flushing the femurs with PBS using one 21G needle. Bone marrow was gently homogenized using a pipette through a 40μm cell strainer. BMCs were counted with a Neubauer plate and suspended in PBS to 1x $10^7$ BMCs/ml. The irradiated recipient *Cse*$^{-/-}$ and *Cse*$^{+/+}$ mice were then injected 1x $10^6$ BMCs in 100 μl volume intravenously via a tail vein. Mice were allowed to recover for 1 month post-BMT. Reconstitution was confirmed by analysis of complete blood counts. Then, mice were infected by intraperitoneally injecting 1x$10^8$ cfu/mouse of MDR *P. aeruginosa* isolate 6–11–19. Survival was recorded for 7 days.

**Blood and serum analysis.** Complete blood count was performed with the ADVIA 2120i system (SIEMENS, Munich, Germany). Creatinine was measured via the Jaffe method, with lowest detection limit 0.03 mg/dl. Aspartate transaminase (AST) and alanine transaminase (ALT) were measured by the IFCC method with the ADVIA 1800 system (SIEMENS). Lowest detection limits were 0.8 U/l and 0.6 UI/l respectively.

**Cytokine stimulation and measurements.** Segments of mice spleens stored in RPMI 1640 were gently squeezed and passed through a sterile filter (250 mm, 12–13 cm, AlterChem Co, Athens, Greece) for the collection of splenocytes. After three serial washings, cells were counted on Neubauer plates with trypan blue for exclusion of dead cells. A total of 5 x $10^6$ cells/ml were then incubated into sterile 24-well plates in RPMI-1640, supplemented with 2 mM glutamine, 10% fetal bovine serum, 100 U/ml of penicillin G and 0.1 mg/ml of streptomycin in the absence or presence of 10 ng/ml LPS of *E. coli* O55:B5 or 5 x $10^5$ cfu/ml heat killed *Candida αlbicans*. After 24 hours or 5 days of incubation at 37˚C in 5% $CO_2$, the plates were centrifuged, and the supernatants were collected. Concentrations of TNFα was measured in duplicate in supernatants from stimulated splenocytes, in tissue supernatants and in serum via enzyme- linked immunosorbent assays (ThermoFisher Scientific, California, USA). The lower detection limit was 39 pg/ml. Additionally, IL-1α, IL-1β, IL-6, IL-10, IL-12, IL-27, IFNγ and IFNβ in serum were determined by the LEGENDplex mouse TH 1/2 cytokine panel (13-plex) (Biolegend, California, USA) according to the manufacturer's instructions.

**Determination of myeloperoxidase (MPO) activity.** Tissue segments were homogenized with T-PER (ThermoFisher Scientific, Massachusetts, USA) and centrifuged at 10,000 rpm at 4˚C. Then the homogenates were incubated in wells of a 96-well plate at 37˚C with 4.2 mM tetramethylbenzidine (Serva, Heidelberg, Germany), 2.5 mM citrate, 5 mM $NaH_2PO_4$ and 1.18 mM $H_2O_2$ pH 5.0 at a final volume of 150 µl. After 5 minutes the reaction was terminated by adding 50 ml 0.18M $H_2SO_4$. Absorbance was read at 450 nm against blank wells. Results were adjusted for tissue sample protein content on Bradford assay (Sigma-Aldrich) and they were expressed as MPO units/mg protein/g.

**Expression quorum sensing bacterial genes.** RNA was isolated from mice lung segments stored in RNAlater (Qiagen, Hilden, Germany), following the RNeasy mini kit protocol (QIAGEN) according to the manufacturer's recommendations. RNA concentration was measured spectrophotometrically using the absorbance ratio of 260/280 nm. Gel electrophoresis was used to check the purity and integrity of the total RNA. Additionally, the fragmented bacterial RNA was analyzed using the Agilent 2100 Bioanalyzer System (Agilent Technologies, Waldbronn, Germany) following the manufacturer's protocol. 1 µg isolated RNA from *P. aeruginosa* was used as template, with the first strand iScript cDNA Synthesis Kit (Bio-Rad, Hercules, CA, USA) for the reverse transcription for cDNA synthesis; conditions were 5 min at 25˚C, 30 min at 42˚C, and 5 min at 85˚C. Quantitative real-time PCR (qRT-PCR) analysis of the expression of quorum sensing (QS) genes *rhll*, *rhlR*, *lasl*, *lasR*, *pqsA* and *pqsR* was carried out using the *P. aeruginosa* specific primers described [24] (Table C in S1 Text). All primers were purchased from IDT (IDT, San Jose, California, U.S.A). qRT- PCR was performed with the iQ5 cycler system (BioRad, Hercules, CA, USA) using 2 µl cDNA, 10 µl iTaq Universal SYBR Green Supermix, (BioRad), 6µl Molecular Grade Water (AppliChem) and 0.1 pmol/ml sense and antisense primers to a final volume of 20-µl on duplicate samples in 96-well plate. The thermal cycler profile involved a preliminary denaturation at 95˚C for 1 min; followed by 40 cycles of denaturation at 95˚C for 30 s; annealing at 52˚C for 30 s; and extension for at 72˚C 1 min.; followed by cooling at 4˚C. Amplification was followed by a melting curve from 55˚C to 95˚C increasing of 0.5 each time for verification of the amplified product. The threshold was adjusted according to the amplification curves of all evaluated genes. No reverse transcriptase controls were prepared from total RNA and no template controls were prepared with Molecular Grade Water (AppliChem) in place of total RNA to indicate potential genomic DNA contamination in isolated bacterial RNA and contamination of reagents, respectively. Specificity was confirmed by electrophoretic analysis in 3% w/v agarose electrophoresis gel (5V/cm) and ethidium bromide staining (AppliChem). Analysis of relative gene expression was achieved according to the $^{\Delta\Delta}$ CT method [25]. The number of transcripts of QS genes in the lung was normalized to the respective number of bacteria in the lung at the time of sacrifice [26].

Additionally, each MDR *P. aeruginosa* isolate was incubated at a starting inoculum of 1 x $10^7$ cfu/ml in tubes with Mueller-Hinton broth with and without 1mM of the $H_2S$ donor GYY4137 (Sigma-Aldrich) and with and without 1mM of the AOAA (Sigma-Aldrich). After 2 and 6 hours of incubation RNA was isolated from 1ml of each dilution, using the same methodology as described above. RNA was then used for measurement of expression of quorum sensing genes. An aliquot of 0.1ml of each dilution was diluted 1:10 into Mueller-Hinton broth (Becton Dickinson) six consecutive times; 0.1 ml of each dilution was plated onto MacConkey agar (Becton Dickinson). After incubation for 24 hours at 37˚C, the number of viable colonies was counted. The results were expressed as $log_{10}$ of colony forming units per ml (cfu/ml). The number of transcripts of QS genes was normalized to the respective bacterial load. The experiment was performed in duplicate.

**Determination of splenic leukocyte subpopulations by flow cytometry.** Isolated spleen cells were incubated for 15 min in the dark with the monoclonal antibodies anti-CD45 PC5

(ThermoFisher Scientific), anti-CD11b PE (phycoerythrin, emission 575nm, ThermoFisher Scientific), anti-CD11c FITC (fluorescein isothiocyanate, emission 525 nm, ThermoFisher Scientific). Cells were analyzed after running through the CYTOMICS FC500 flow cytometer (Beckman Coulter Co, Miami, Florida). Results were expressed as percentages.

**Bacterial outgrowth assays. i)** *in vitro* **clearance.** A log-phase culture ($1 \times 10^5$ cfu/ml) of MDR *P. aeruginosa* was opsonized with 5% mouse serum (Sigma-Aldrich) at room temperature for 15 min. $5 \times 10^6$ leukocytes/ ml were isolated from mice spleens as described above, and were incubated at 37°C in 5% $CO_2$ for 30 min into sterile 24-well plates in RPMI 1640 supplemented with 10% FBS with and without the $H_2S$ donor 1mM GYY3147 (Sigma-Aldrich) [27]. Then, $1 \times 10^5$ cfu/ml bacteria were added in each well and the plates were incubated at 37°C. **ii)** *in vivo.* One aliquot of 0.1 ml of the tissue homogenates was diluted 1:10 into Mueller-Hinton broth six consecutive times; 0.1 ml of each dilution was plated onto MacConkey agar (Becton Dickinson). After incubation for 24 hours at 37°C, the number of viable colonies was counted. The results were expressed as $\log_{10}$ of colony forming units per gram tissue (cfu/g).

## Quantification and statistical analysis

Categorical data were presented as frequencies and quantitative variables as mean ± 95% confidence intervals (CIs) or median + interquartile range (IQR). Comparisons between groups were done using the Fisher exact test (categorical data), and the Mann-Whitney U test for two group comparisons, the Wilcoxon signed rank test for related samples comparison and one-way ANOVA with the Bonferroni correction for multiple group comparison (quantitative data). Correlations between variables were performed using the Spearman's rank of order. Resolution of VAP in patients and survival in mice was compared between groups by the log-rank test. Odds ratios (OR) and 95% confidence intervals (CIs) for were calculated by the Mantel and Haenszel's statistics. The receiver operating characteristic (ROC) curve was analyzed for the efficiency of serum hydrogen sulfide as a biomarker for the resolution of VAP. Other demographic variables associated with unfavorable outcome were transformed into dichotomous variables after ROC curve analysis. For each parameter the coordinate point with the maximum value of the Youden index was used as a cut- off. Step- wise Logistic regression analysis with odds ratios (ORs) and confidence intervals (CIs) was used to investigate if serum hydrogen sulfide is an independent variable for resolution of VAP. Any *p* value below 0.05 was considered statistically significant.

## Results

### Clinical study

The clinical study flow chart is shown in S1 Fig. We used a dataset of 6814 patients with sepsis from Greek hospitals. The multi- step selection process involved the following filters: a) selection of patients with VAP-related sepsis; b) selection of patients with microbiological confirmation of VAP by a single pathogen (pathogen growth in TBS $\geq 10^5$ cfu/ml and exclusion of patients growing more than one microorganism); and c) selection of patients with monomicrobial infection by *P. aeruginosa*, *K. pneumoniae* or *A. baumannii* as defined by the isolation of these species in tracheobronchial secretions. Following this approach, we ended-up with 225 patients who developed sepsis due to microbiologically confirmed VAP by *P. aeruginosa* (71 patients; 31.5%), *K. pneumoniae* 60 patients, 26.7%) and *A. baumannii* (94 patients,41.8%), respectively. Demographics of groups were similar (Table A in S1 Text).

Bacterial VAP led to reduced serum $H_2S$ levels compared to healthy volunteers (S2 Fig). When serum levels of $H_2S$ on day 1 of enrollment were compared between 28-day survivors

and non-survivors, they were higher only among patients with infection by *P. aeruginosa* (p < 0.0001) (Fig 1A). Compared to healthy volunteers, serum $H_2S$ levels of survivors of infection by *P. aeruginosa* did not differ significantly, whereas non-survivors had significantly lower $H_2S$ levels. These findings suggest that defective host- derived $H_2S$ responses may be specific for unfavorable outcome of *P. aeruginosa* infections and prompted further study on patients with *P. aeruginosa* infection.

Following ROC curve analysis, it was found that serum levels of $H_2S$ on day 1 above 5.3μM had the best trade-off for sensitivity and specificity for survival from sepsis due to VAP by *P. aeruginosa* (Fig 1B and 1C). In total, 55 patients had higher and 16 patients had less than or equal to 5.3 μM $H_2S$ in serum on day 1. After 28 days 38 (69.1%) patients with levels > 5.3 μM and 1 (6.3%) patient with levels ≤ 5.3 μM survived (Fig 1D). The odds ratio (OR) for survival from sepsis due to VAP by *P. aeruginosa* was 33.33 (95% Confidence Interval (CI): 4.10–250.00, p = 7 x $10^{-6}$).

ROC curve analysis revealed the following baseline values to be associated with unfavourable outcome: age > 61 years, Acute Physiology and Chronic Health Evaluation (APACHE) II score > 23, Charlson Comorbidity Index (CCI) > 4, Sequential Organ Failure Assessment (SOFA) score on day 1 > 10 and bacteremia by *P. aeruginosa*. All above variables entered into logistic regression analysis (Table 1). Analysis showed that serum $H_2S$ on day 1 above 5.3 μM was an independent protective factor for favorable outcome.

## Animal studies

Following the results of the human study, we aimed to assess the potential protective role of host derived $H_2S$ and investigated the underlying mechanisms in an animal model of severe MDR *P. aeruginosa* infection. 7-day survival of *Cse*$^{-/-}$ mice was significantly lower (11.8%) as compared to *Cse*$^{+/+}$ mice (47.1%) (Fig 2A). This finding could be replicated after experimental infection with two additional MDR P. aeruginosa strains (S3B and S3C Fig). Application of STS and GYY4137, two clinically used $H_2S$ donors, restored the survival of *Cse*$^{-/-}$ mice to a level comparable to wild-type mice (43.3% and 46.7% respectively compared to 11.8% of untreated *Cse*$^{-/-}$ mice) (Fig 2B). This was further confirmed using a pharmacological approach. In *Cse*$^{+/+}$ mice infected by *P.aeruginosa*, the $H_2S$ biosynthesis inhibitor AOAA significantly decreased survival (11.8% compared to 47.1% of untreated *Cse*$^{+/+}$ mice) (Fig 2C). The selective CSE inhibitor PAG also significantly decreased survival (11.8% compared to 47.1% of untreated *Cse*$^{+/+}$ mice) (Fig 2C). Treatment with STS in *Cse*$^{+/+}$ mice led to prolongation of survival (64.7% compared to 47.1%). MDR *P. aeruginosa* infection led more rapidly to the development of multiorgan failure (MOF) in the *Cse*$^{-/-}$ mice compared with the *Cse*$^{+/+}$ mice, as shown by the greater increase of serum aminotransferases ALT and AST and serum creatinine (Fig 2D–2F).

We next investigated if the protective role of host-derived $H_2S$ is enzyme- and pathogen-specific. First, we demonstrated that $H_2S$ levels measured in serum and tissue samples from non- infected and infected by MDR *P. aeruginosa* mice with a physiological expression of *Cse* (*Cse*$^{+/+}$) were significantly higher compared to non- infected and infected mice with a constitutive knock-out of the gene encoding for Cse (*Cse*$^{-/-}$) respectively (Fig 3A–3C), showing that CSE deletion leads to deficient $H_2S$ levels in vivo. Moreover, $H_2S$ levels in serum increased after infection by MDR *P. aeruginosa*, only in *Cse*$^{+/+}$ mice. In order to study the importance of the enzymatic system of host $H_2S$ production, we performed another set of experiments using mice with homozygous deficiency for *3Mst*. *3Mst* deletion does not lead to a significant alteration of serum $H_2S$ levels in mice [28]. Indeed, survival from infection by MDR *P. aeruginosa* was similar between *3Mst*$^{+/+}$ (42.9%) and *3Mst*$^{-/-}$ (38.5%) mice (Fig 3D). These findings

A.

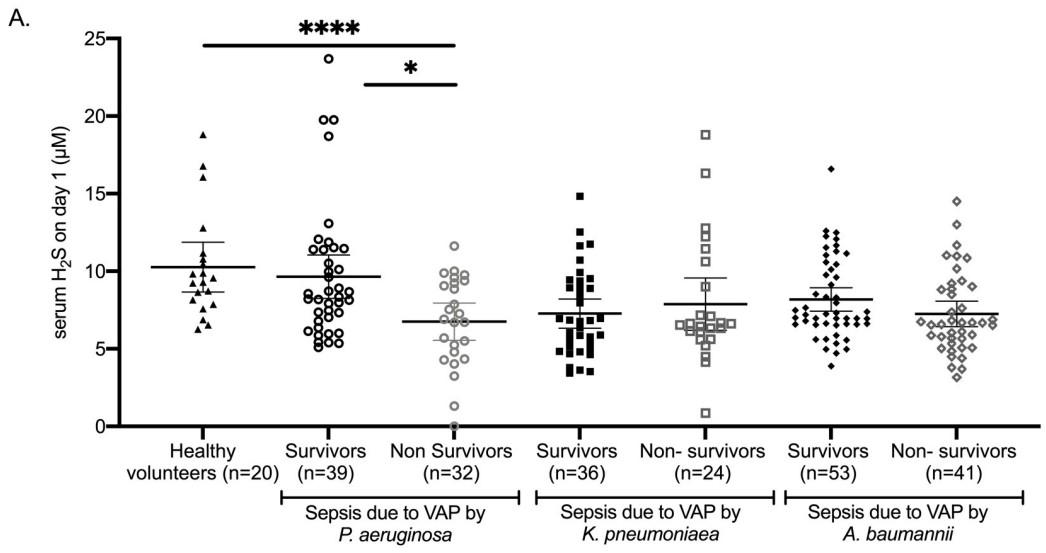

B.

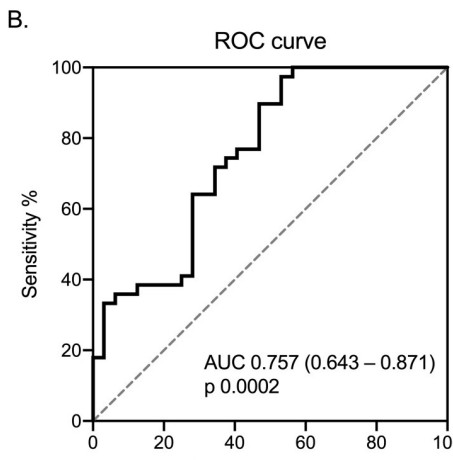

C.

| | Survivors (n patients) | Non- Survivors (n patients) | Total |
|---|---|---|---|
| H$_2$S > 5.3µM | 36<br>Sensitivity: 92.3%<br>PPV: 66.7%% | 18 | 54 |
| H$_2$S ≤ 5.3µM | 3 | 14<br>Specificity: 43.8%<br>NPV: 82.4% | 17 |
| | 39 | 32 | 71 |

D.

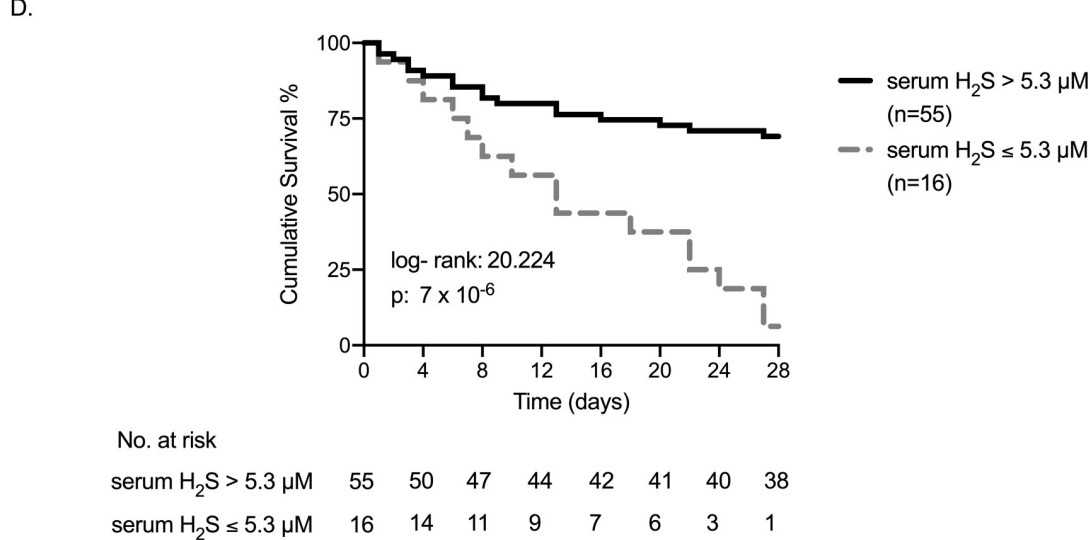

**Fig 1. Circulating H$_2$S and outcome of patients with sepsis due to ventilator associated pneumonia (VAP) by *P. aeruginosa*.** A) Serum levels of H$_2$S in healthy volunteers and on day 1 among survivors and non survivors of sepsis due to VAP by three major pathogens were measured by HPLC. Only statistically significant comparisons are provided (comparisons by the Mann Whitney U test); * p< 0.05, **** p< 0.0001. B) ROC curve of H$_2$S to predict survival of sepsis due to VAP by *P. aeruginosa*, AUC area under the curve. C) Sensitivity, specificity, positive predictive value (PPV), and negative predictive value (NPV) of serum H$_2$S level >5.3 µM for survival of patients with sepsis due to VAP by *P. aeruginosa*. D) Kaplan- Meier analysis for survival after sepsis due to VAP by *P. aeruginosa*, among patients with serum H$_2$S levels > and ≤ 5.3 µM. Results of the log-rank test and the p- value are given.

demonstrate that the susceptibility of the host for infection by *P. aeruginosa* is selectively driven by H$_2$S produced through the catalytic activity of CSE.

Similar to prior findings [29], the difference in MDR *P. aeruginosa* virulence in *Cse*$^{-/-}$ and *Cse*$^{+/+}$ mice is not related to LPS, as there was no difference in survival of *Cse*$^{-/-}$ and *Cse*$^{+/+}$ mice challenged either with 10 mg/kg/mouse LPS or 30 mg/kg/mouse LPS (Fig 3E and 3F). The protective role of H$_2$S could not be reproduced after KPC-*K. pneumoniae* infection (47.6% survival of both *Cse*$^{-/-}$ and *Cse*$^{+/+}$ mice) (Fig 3G), indicating a pathogen-specific effect.

We then sought to investigate the mechanisms that lead to death in the case of host H$_2$S deficiency. Cytokine analysis did not show significant differences between *Cse*$^{-/-}$ and *Cse*$^{+/+}$ mice in TNFα levels in serum and tissue supernatants and in stimulated splenocytes (S4 Fig). Interleukin (IL)-1α, IL-1β, IL-6, IL-10, IL-12, IL-27, Interferon (IFN)γ and IFNβ levels of serum were also similar between groups (S5 Fig). This suggests that the protective role is not directly associated with a change in the concentration in the blood of circulating cytokines.

We next investigated if the protective effect of H$_2$S stems from favorable phagocytosis. To investigate this, we followed a multistep approach. At first, we verified that absolute counts of

**Table 1. Baseline and clinical characteristics of patients with sepsis due to *P. aeruginosa* associated VAP, univariate and step- wise forward logistic regression analysis of parameters associated with unfavorable outcome of sepsis due to VAP by *P. aeruginosa*.**

| | Survivors | Non Survivors | Univariate analysis | | Step- wise logistic regression analysis | |
|---|---|---|---|---|---|---|
| | (n = 39) | (n = 32) | OR (95% CIs) | p | OR (95% CIs) | p |
| Male gender (n, %) | 29 (74.4) | 18 (56.3) | 0.44 (0.16–1.21) | 0.134 | | |
| Age > 61 (n, %) | 20 (51.3) | 24 (75) | 2.85 (1.03–7.88) | 0.035 | | |
| APACHE II 1 > 23 (n, %) * | 4 (10.3) | 14 (43.8) | 6.81 (1.95–23.71) | 0.002 | | |
| CCI > 4 (n, %) * | 8 (20.5) | 17 (53.1) | 4.39 (1.55–12.45) | 0.006 | | |
| SOFA day 1 > 10 (n, %) * | 5 (12.8) | 18 (56.3) | 8.74 (2.71–28.17) | 0.0001 | 16.16 (4.00–65.23) | 0.00009 |
| H$_2$S in serum on day 1 > 5.3 µM *(n, %) | 38 (97.4) | 17 (53.1) | 0.03 (0.01–0.24) | 7x10$^{-6}$ | 0.02 (0.02–0.15) | 0.0003 |
| Septic shock (n, %) | 22 (56.4) | 25 (78.1) | 2.76 (0.97–7.89) | 0.078 | | |
| Pathogen isolation in a blood sample (n%) | 6 (15.4) | 12 (37.5) | 3.3 (1.07–10.18) | 0.032 | | |
| Appropriateness of antimicrobial therapy (n%) | 25 (64.1) | 21 (65.6) | 1.07 (0.40–2.85) | 0.547 | | |
| Intake of corticosteroids (n%) | 4 (10.3) | 7 (21.9) | 2.45 (0.65–9.28) | 0.204 | | |
| Main comorbidities (n%) | | | | | | |
| • Diabetes mellitus Typ 2 | 4 (10.3) | 2 (6.3) | 0.58 (0.10–3.41) | 0.683 | | |
| • Chronic heart failure | 2 (5.1) | 5 (15.6) | 3.43 (0.62–19.00) | 0.231 | | |
| • Coronary heart disease | 8 (20.5) | 2 (6.3) | 0.26 (0.51–1.32) | 0.102 | | |
| • COPD | 3 (7.7) | 3 (9.4) | 1.24 (0.23–6.62) | 0.564 | | |
| • Chronic renal failure | 1 (2.6) | 0 (0) | 0.97 (0.93–1.03) | 0.549 | | |
| • Solid tumor | 1 (2.6) | 1 (3.1) | 1.23 (0.74–20.40) | 0.702 | | |

[A]Abbreviations VAP: Ventilator associated pneumonia; APACHE: Acute physiology and chronic health evaluation; CCI: Charlson's Comorbidity Index; SOFA: Sequential organ failure assessment; H$_2$S: hydrogen sulfide; OR: Odds ratio; CI: Confidence intervals; COPD: Chronic obstructive pulmonary disorder

[B]#Cut- off point of each variable was determined based on the coordinate point with the maximum value of the Youden index

A.

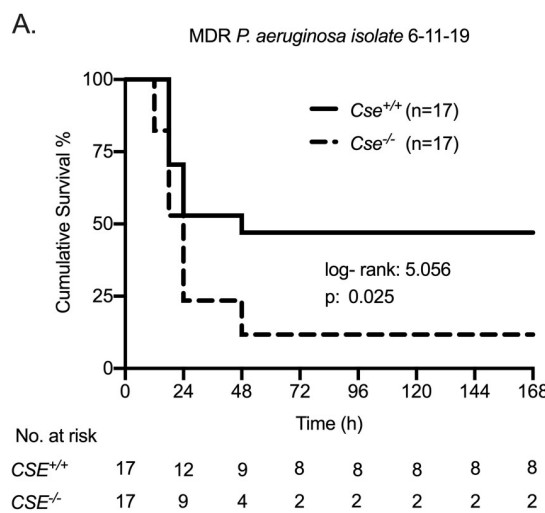

MDR *P. aeruginosa isolate* 6-11-19

*Cse+/+* (n=17)
*Cse-/-* (n=17)

log- rank: 5.056
p: 0.025

No. at risk

| | | | | | | | | |
|---|---|---|---|---|---|---|---|---|
| *CSE+/+* | 17 | 12 | 9 | 8 | 8 | 8 | 8 | 8 |
| *CSE-/-* | 17 | 9 | 4 | 2 | 2 | 2 | 2 | 2 |

B.

C.

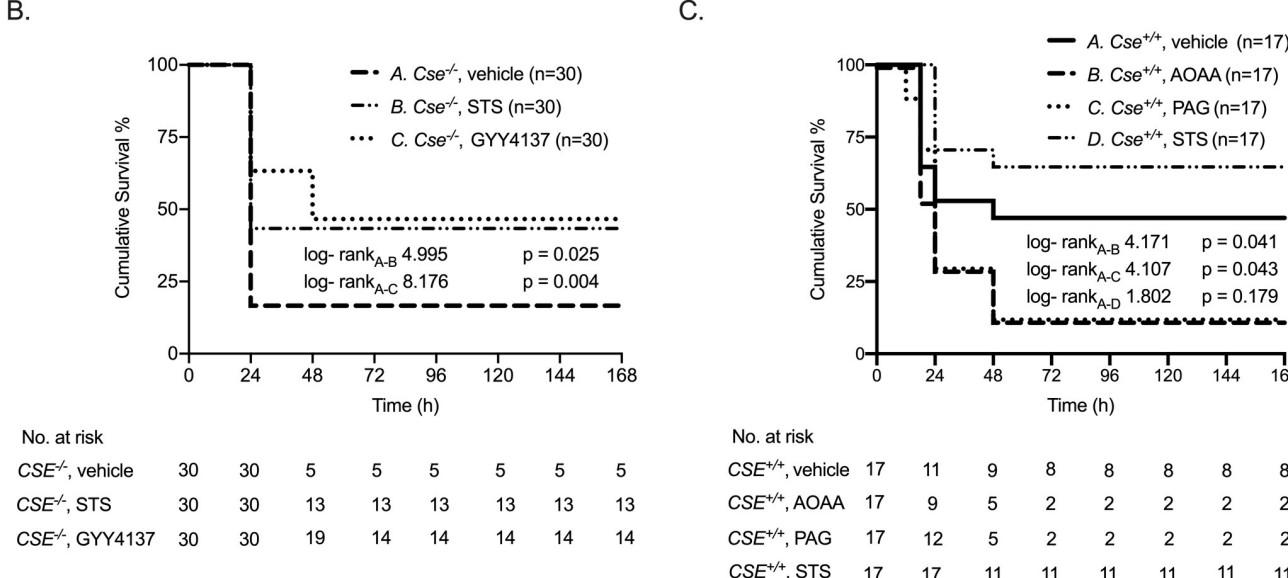

*A. Cse-/-*, vehicle (n=30)
*B. Cse-/-*, STS (n=30)
*C. Cse-/-*, GYY4137 (n=30)

log- rank$_{A-B}$ 4.995        p = 0.025
log- rank$_{A-C}$ 8.176        p = 0.004

*A. Cse+/+*, vehicle  (n=17)
*B. Cse+/+*, AOAA (n=17)
*C. Cse+/+*, PAG (n=17)
*D. Cse+/+*, STS (n=17)

log- rank$_{A-B}$ 4.171    p = 0.041
log- rank$_{A-C}$ 4.107    p = 0.043
log- rank$_{A-D}$ 1.802    p = 0.179

No. at risk

| | | | | | | | | |
|---|---|---|---|---|---|---|---|---|
| *CSE-/-*, vehicle | 30 | 30 | 5 | 5 | 5 | 5 | 5 | 5 |
| *CSE-/-*, STS | 30 | 30 | 13 | 13 | 13 | 13 | 13 | 13 |
| *CSE-/-*, GYY4137 | 30 | 30 | 19 | 14 | 14 | 14 | 14 | 14 |

No. at risk

| | | | | | | | | |
|---|---|---|---|---|---|---|---|---|
| *CSE+/+*, vehicle | 17 | 11 | 9 | 8 | 8 | 8 | 8 | 8 |
| *CSE+/+*, AOAA | 17 | 9 | 5 | 2 | 2 | 2 | 2 | 2 |
| *CSE+/+*, PAG | 17 | 12 | 5 | 2 | 2 | 2 | 2 | 2 |
| *CSE+/+*, STS | 17 | 17 | 11 | 11 | 11 | 11 | 11 | 11 |

D.

E.

F.

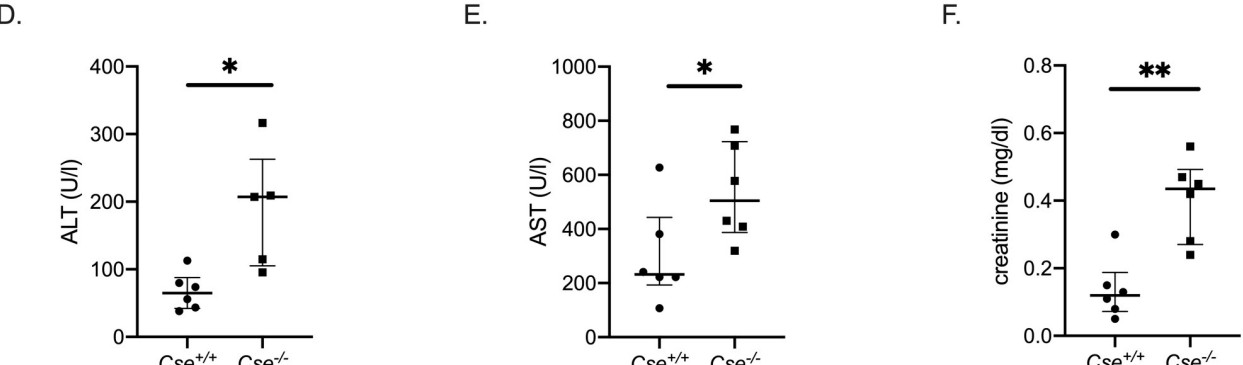

**Fig 2. Effect of H$_2$S production deficiency on the outcome of infection by *P. aeruginosa*.** Survival analysis between A) *Cse*$^{+/+}$ and *Cse*$^{-/-}$ mice after infection with MDR *P. aeruginosa* (isolate 6–11–19); B) *Cse*$^{-/-}$ mice after infection and treatment with vehicle or the H$_2$S- donor sodium thiosulfate (STS) and the H$_2$S-donor GYY4137, C) *Cse*$^{+/+}$ mice after infection and treatment with vehicle or the H$_2$S synthesis inhibitor aminooxyacetic acid (AAA) or the selective CSE inhibitor Propargylglycine (PAG) or STS. Results of the log-rank test and the relevant p- values are given. D-F) Mean concentration of alanine transaminase (ALT) and aspartate transaminase (AST) and creatinine in serum of *Cse*$^{+/+}$ and *Cse*$^{-/-}$ mice (n = 6) 6 hours after experimental infection. Comparisons by the Mann Whitney U test; $^*$ p< 0.05, $^{**}$ p< 0.01.

all blood cell lines, including neutrophils and lymphocytes, did not differ between healthy *Cse*$^{+/+}$ and *Cse*$^{-/-}$ mice (S6 Fig). Then, we studied mice rendered neutropenic by cyclophosphamide and we found that the survival offered by *Cse*$^{+/+}$ was lost: neutropenic *Cse*$^{-/-}$ mice and neutropenic *Cse*$^{+/+}$ mice exhibited similarly high mortality rates (Fig 4A). In a different set of experiments, survival was assessed in *Cse*$^{-/-}$ and *Cse*$^{+/+}$ mice after BMT from a *Cse*$^{+/+}$ donor. Although BMT did not significantly alter serum hydrogen levels of *Cse*$^{-/-}$ and *Cse*$^{+/+}$ mice (S7 Fig), transplanted *Cse*$^{-/-}$ mice had prolonged survival compared to naïve *Cse*$^{-/-}$ mice, which was similar to the survival of naïve *Cse*$^{+/+}$ mice and *Cse*$^{+/+}$ mice after BMT (Fig 4B). Taken together, these findings suggest that the survival benefit of *Cse*$^{+/+}$ over *Cse*$^{-/-}$ mice is not owed to a baseline excess of immune cells, but potentially to an intrinsic dysfunction of *Cse*$^{-/-}$ neutrophils, which is reverted after BMT. Additionally, inflammatory mediators produced by neutrophils, namely IL-23, MCP-1 and GM-CSF were lower in *Cse*$^{-/-}$ mice compared to *Cse*$^{+/+}$ mice (Fig 4C). Moreover, the expression of these mediators increased after infection in *Cse*$^{+/+}$ mice. In addition, the bacterial load of all three MDR *P. aeruginosa* isolates 6 h after infection was greater in the liver, in the lung and in the spleen of *Cse*$^{-/-}$ mice (Figs 4D, S2D and S2E). Myeloperoxidase (MPO) activity, which is a marker for the presence of neutrophils in tissue homogenates, was lower in the liver and in the lung of *Cse*$^{-/-}$ mice compared with *Cse*$^{+/+}$ mice 6h after *P. aeruginosa* infection (Fig 4E). The ratio of CD11b$^+$CD11c$^-$CD45 cells to CD11b$^-$CD11c$^+$CD45 of the spleen, which mostly represent the ratio of the granulocyte to the macrophage population *[30]*, was significantly greater in *Cse*$^{+/+}$ mice compared to *Cse*$^{-/-}$ mice 6h after *P.aeruginosa* infection (Fig 4F). A negative correlation between neutrophil to macrophages ratio and bacterial load could be shown, suggesting that greater counts of neutrophils are associated with an effective clearance of MDR *P. aeruginosa* (Fig 4G). We then sought to investigate the influence of H$_2$S on the *in vitro* phagocytic activity of leukocytes. Indeed, we found that phagocytic activity of leukocytes was lower in *Cse*$^{-/-}$ mice than in *Cse*$^{+/+}$ mice. Phagocytotic activity of leukocytes from *Cse*$^{-/-}$ mice was normalized after preincubation with the H$_2$S donor GYY4137. GYY4137 enhanced also the phagocytic ability of *Cse*$^{+/+}$ derived leucocytes after 2 hours but not after 4 hours from bacterial challenge. (Fig 4H and 4I). All the above findings suggest a deficient mechanism of neutrophil recruitment and neutrophil mediated phagocytosis in *Cse*$^{-/-}$ mice. This is further proved by the positive correlation between bacterial load and tissue MPO activity in tissues of *Cse*$^{+/+}$ mice, which was not seen in *Cse*$^{-/-}$ mice (S8 Fig).

The fact that endogenous H$_2$S, produced from the CSE enzyme, exerts a protective mechanism against MDR *P. aeruginosa* infection but not against infection elicited by other gram-negative bacteria, such as *K. pneumoniae*, indicates that host-derived H$_2$S interacts with a unique system that is crucial for the pathogenesis of infections by *P. aeruginosa*. Quorum sensing (QS) is such a unique system, which controls the virulence of *P. aeruginosa* *[24,31]*. Modulation of H$_2$S levels, by the addition of an H$_2$S donor or an H$_2$S biosynthesis inhibitor, did not affect *in vitro* bacterial growth (Fig 5A). Then, we studied the expression of QS genes of the three selected isolates under different concentrations of H$_2$S. Increased H$_2$S levels, through the addition of an H$_2$S donor, led to a reduction of expression of core QS genes over time compared to normal or reduced H$_2$S levels (Fig 5B–5F). Then, we investigated the effect of tissue

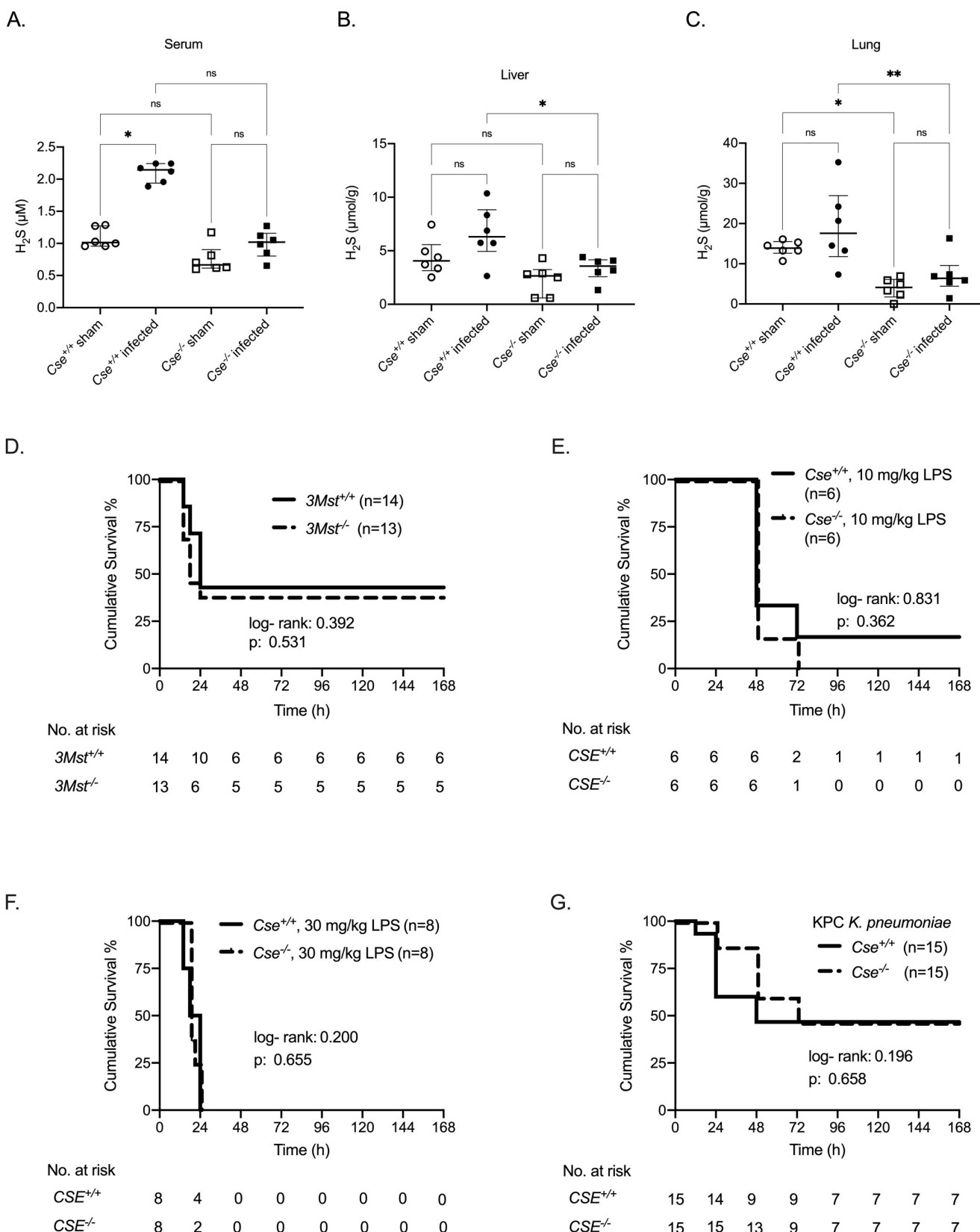

**Fig 3. Specific effect of CSE enzyme on H$_2$S production and the outcome of multi-drug resistant (MDR) *Pseudomonas aeruginosa* experimental infection.** Concentration of H$_2$S in A) serum; B) liver; and C) lung from non- infected $Cse^{+/+}$ and $Cse^{-/-}$ mice and from $Cse^{+/+}$ and $Cse^{-/-}$ mice 6 hours after experimental infection (n = 6 per group). Comparison by the ANOVA test with Bonferroni correction for multiple comparisons; ns non-significant, $^*$ p< 0.05, $^{**}$ p< 0.01, $^{****}$ p< 0.0001. Survival analysis between: D) $3Mst^{+/+}$ and $3Mst^{-/-}$ after experimental MDR *P. aeruginosa* (isolate 6–11–19) infection, E) $Cse^{+/+}$ and $Cse^{-/-}$ mice after i.p. injection of 10 mg/kg LPS, F) $Cse^{+/+}$ and $Cse^{-/-}$ mice after i.p. injection of 30 mg/kg LPS, G) $Cse^{+/+}$ and $Cse^{-/-}$ after experimental KPC (+) *K. pneumoniae* infection. Results of the log-rank test and the relevant p- values are given.

H$_2$S levels on QS expression. Indeed, the expression of QS genes *rhll*, *lasR* and *pqsA* in the lung was decreased over time in $Cse^{+/+}$ but not in $Cse^{-/-}$ mice (Fig 6A–6F), and after 24 hours expression of QS genes *rhll*, lasl and *lasR* in the lung was significantly greater in $Cse^{-/-}$ mice compared to $Cse^{+/+}$ mice. These findings suggest a H$_2$S mediated modulation of QS activation. Moreover, we found a negative correlation between the expression of QS gene *rhll*, *rhlR*, *lasl*, *lasR* and *pqsA* in the lung and the H$_2$S levels in the lung of $Cse^{+/+}$ mice. This was not seen in $Cse^{-/-}$ mice (S9 Fig). These results imply a central role of H$_2$S produced specifically through CSE in the modulation of the QS system, which regulates the tissue outgrowth of MDR *P. aeruginosa*.

## Discussion

Our study adds novel clinical data about the importance of host- derived H$_2$S for the pathogenesis of infection by *P. aeruginosa*. Our conclusions are supported both by clinical observations in humans and by mechanistic data in mice. Non-survivors from severe infection by *P. aeruginosa* fail to produce as high levels of H$_2$S as survivors. This finding is fully corroborated in animal findings where the presence of H$_2$S protects against *P. aeruginosa* induced mortality through a dual mechanism of action: it primes phagocytosis by host neutrophils; and it is a negative regulator of QS that is crucial in the pathogenesis of *P. aeruginosa* infection in the lung.

Hydrogen sulfide production through the CSE enzyme has been observed both in pro- and eucaryotic organisms. This evolutionary conservation of CSE derived H$_2$S, together with findings from several studies, which have demonstrated H$_2$S as a mediator of several human physiological functions, longevity and stress resistance, have shifted the point of view of the research community away from considering H$_2$S as a toxic gas and recognizing its place as a third gaseous transmitter [3,5,32].

Neutrophils, by intracellular killing following phagocytosis, are considered to be the major mediators of the innate immune response in the case of an acute *P. aeruginosa* infection [33,34]. But also, in the case of chronic infection, such as in cystic fibrosis (CF), neutrophilic activity is a key mediator for clearance of *P. aeruginosa*, mainly by the formation of neutrophil extracellular traps (NETs) [35,36]. The key role of neutrophils for the survival advantage of mice with a physiological expression of $Cse^{+/+}$ over $Cse^{-/-}$ mice, is supported by the fact that this advantage is eliminated when neutrophils are depleted. Additionally, hematopoietic reconstitution of the bone marrow of $Cse^{-/-}$ mice through transplantation of bone marrow cells from $Cse^{+/+}$ mice led to a significant increase of survival after *P. aeruginosa* infection. Moreover, $Cse^{+/+}$ mice have higher levels of MPO which is a surrogate marker of neutrophil recruitment and neutrophil activity. Production of MCP-1, Il-23 and GM-CSF, inflammatory mediators which regulate the chemotaxis and activity of neutrophils chemotaxis, increased after infection but only in $Cse^{+/+}$ mice suggesting a defect in chemotaxis of neutrophils induced by the lack of H$_2$S. The a) decreased efficiency of leukocytes of $Cse^{-/-}$ mice in killing *P. aeruginosa in vitro*; b) the reversal of this phenomenon after restoration of H$_2$S levels through pre-incubation with an H$_2$S donor; and c) the

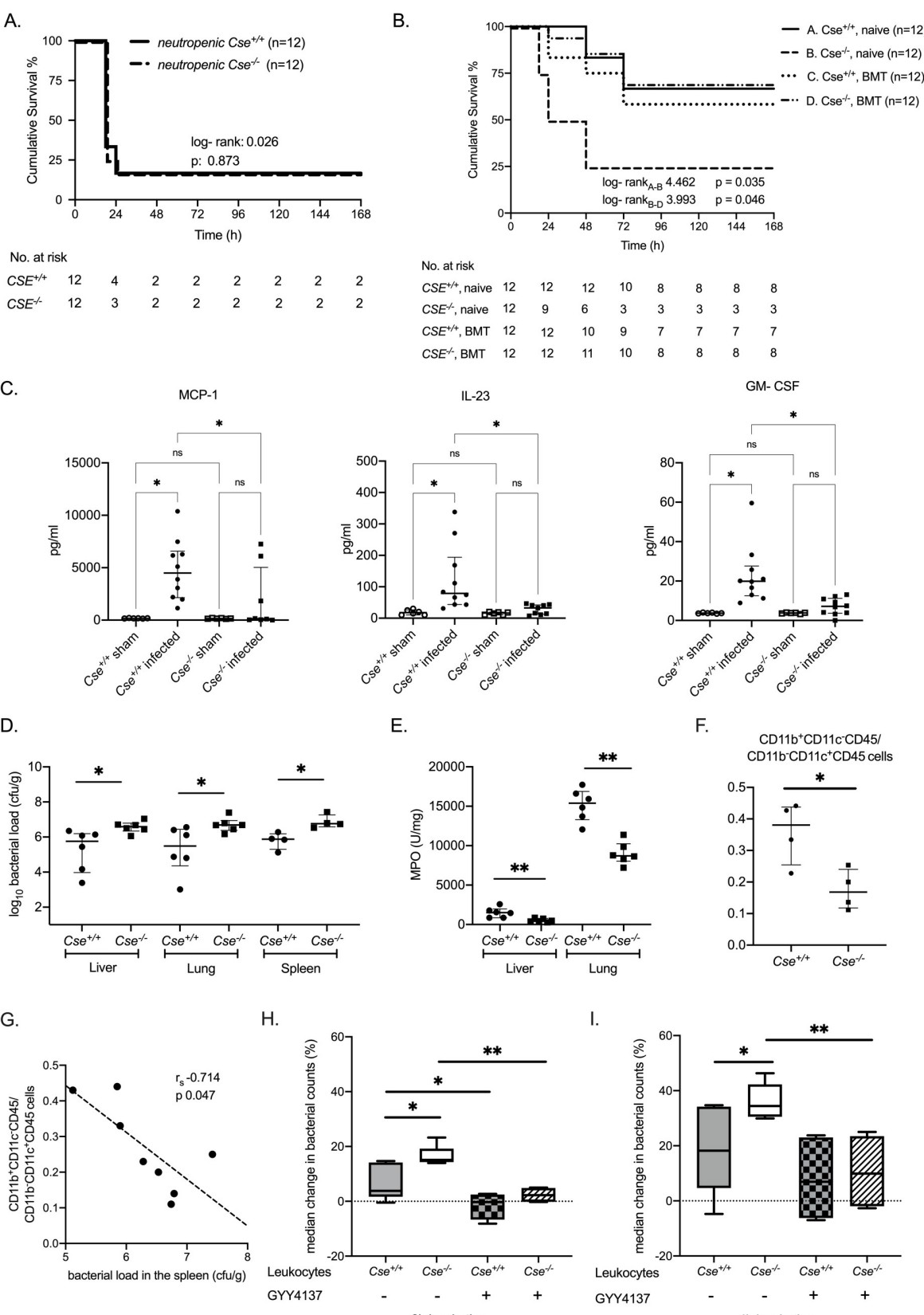

**Fig 4. Effect of CSE enzyme on the phagocytosis of multi-drug resistant (MDR) *Pseudomonas aeruginosa*.** A) Survival analysis of neutropenic $Cse^{+/+}$ and $Cse^{-/-}$ mice after experimental MDR *P. aeruginosa* (isolate 6–11–19) infection. Results of the log-rank test and relevant p- value are given. B) Survival analysis after experimental MDR *P. aeruginosa* (isolate 6–11–19) infection of naive $Cse^{+/+}$ and $Cse^{-/-}$ mice and $Cse^{+/+}$ and $Cse^{-/-}$ after bone marrow transplantation (BMT). Results of the log-rank test and relevant p- values are provided. C- F) $Cse^{+/+}$ and $Cse^{-/-}$ mice were sacrificed 6h after experimental infection by MDR *P. aeruginosa* isolate 6–11–19. C) Serum levels of IL-23, MCP-1 and GM- CSF; Levels of non- infected $Cse^{+/+}$ and $Cse^{-/-}$ mice are also shown. Comparison by the ANOVA test with Bonferroni correction for multiple comparisons; ns non-significant, $^*$ p< 0.05, $^{**}$ p< 0.01. D) Bacterial load (cfu/g) in the liver, in the lung and in the spleen. Comparisons by the Mann Whitney U test. $^*$ p< 0.05. E) Concentration of MPO (U/mg) in the liver and in the lung. Comparisons by the Mann Whitney U test. $^*$ p< 0.05, $^{**}$ p< 0.01. F) Ratio of CD11b$^+$CD11c$^-$CD45 cells to CD11b$^-$CD11c$^+$CD45 cells in the spleen determined by flow cytometry. Comparison by the Mann Whitney U test. $^*$ p< 0.05. G) Correlation between CD11b$^+$CD11c$^-$CD45 cells to CD11b$^+$CD11c$^-$CD45 cells and bacterial load in the spleen. Spearmann rank correlation coefficient ($r_s$), interpolation line and relevant p- value are given. H-I) $Cse^{+/+}$ and $Cse^{-/-}$ mice (n = 6) were sacrificed and total splenic leukocytes were collected. 5 x 10$^6$ leukocytes/ ml, pre-treated with/ without the H$_2$S donor 1 mM GYY3147, were then incubated with 1 x 10$^5$ cfu/ml of MDR *P. aeruginosa* (isolate 1). Change of bacterial growth compared to baseline was monitored after 2 and 4 hours of incubation. Comparisons by the ANOVA test with Bonferroni correction for multiple comparisons; ns non-significant, $^*$ p< 0.05, $^{**}$ p< 0.01.

increased bacterial loads in infected $Cse^{-/-}$ mice suggest that host H$_2$S is significant for the efficient phagocytosis of *P. aeruginosa*.

*P. aeruginosa* consists one of the most usual causes for nosocomial infections and is characterized by very high resistance rates against 3$^{rd}$ and 4$^{th}$ generation antibiotics in many countries including Greece [1]. Patients with cystic fibrosis (CF) are more susceptible to *P. aeruginosa*. Up to 80% of deaths of patients with CF is attributed to a *P. aeruginosa* infection [31]. The high virulence of *P. aeruginosa* in CF patients relates to its ability to avoid phagocytosis by forming complex biofilms, which help *P. aeruginosa* grow in the patients airway [35]. The biofilm formation remains under control of the QS system, which is unique cell to cell communication mechanism of *P. aeruginosa*. The function of the QS system is regulated by three systems, controlled by the *rhlI / rhlR, lasI / lasR* and *pqsA / pqsR* from which *rhlI, lasI* and *pqsA genes* are responsible for the synthesis of acyl homoserine lactones (AHLs) and *rhlR, lasR* and *pqsR* are transcriptional regulators [24]. AHLs are diffusible signaling molecules. An increase in the concentration of extracellular AHLs, most often as an answer to the increasing number of *P. aeruginosa* colonies, leads to an increase of intracellular AHLs, which boost the expression of genes that regulate the growth of *P. aeruginosa* and the biofilm formation [37]. *P. aeruginosa* cells remain in the biofilm, protected from phagocytosis from tissue macrophages [38]. *In vivo* data show that the inhibition of the expression of one or two of the core QS genes *rhlI* and *lasI* decreases the ability of *P. aeruginosa* to induce an acute pulmonary infection in a murine model [39]. In a prospective clinical study of 60 patients with exacerbation of chronic cystic fibrosis, the concentration of eight different AHLs in the sputum, in plasma and in urine samples was measured. The patients suffered from chronic CF and the measurements were made in the phase of acute exacerbation and also after the administration of antibiotic treatment. The concentrations of the eight AHL molecules were significantly higher at the initial phase of CF exacerbation and were positively correlated with an increase in the *P. aeruginosa* load in sputum. AHL concentration was significantly decreased in the case of successful antibiotic treatment [40]. Moreover, an analysis of sputum samples from 22 children with CF showed that higher concentrations of H$_2$S in sputum were inversely correlated with the amount of sputum produced and was negatively correlated with the likelihood for inpatient hospitalization or oxygen supplementation, suggesting that sulfide could be a marker of good health in CF patients [41]. This effect of hydrogen sulfide was attributed to an enhancement of the reduction capacity, possibly rendering the airway microenvironment inhospitable for aerobes. The findings of our study are consistent with published data since expression of QS genes *rhlI, lasR* and *pqsA* was decreased over time in $Cse^{+/+}$ but not $Cse^{-/-}$ mice. Moreover, we demonstrated that modulation of H$_2$S levels in vitro also influences QS expression of *P. aeruginosa* isolates. Thus, under the presence of H$_2$S *P. aeruginosa* becomes over time less efficient in

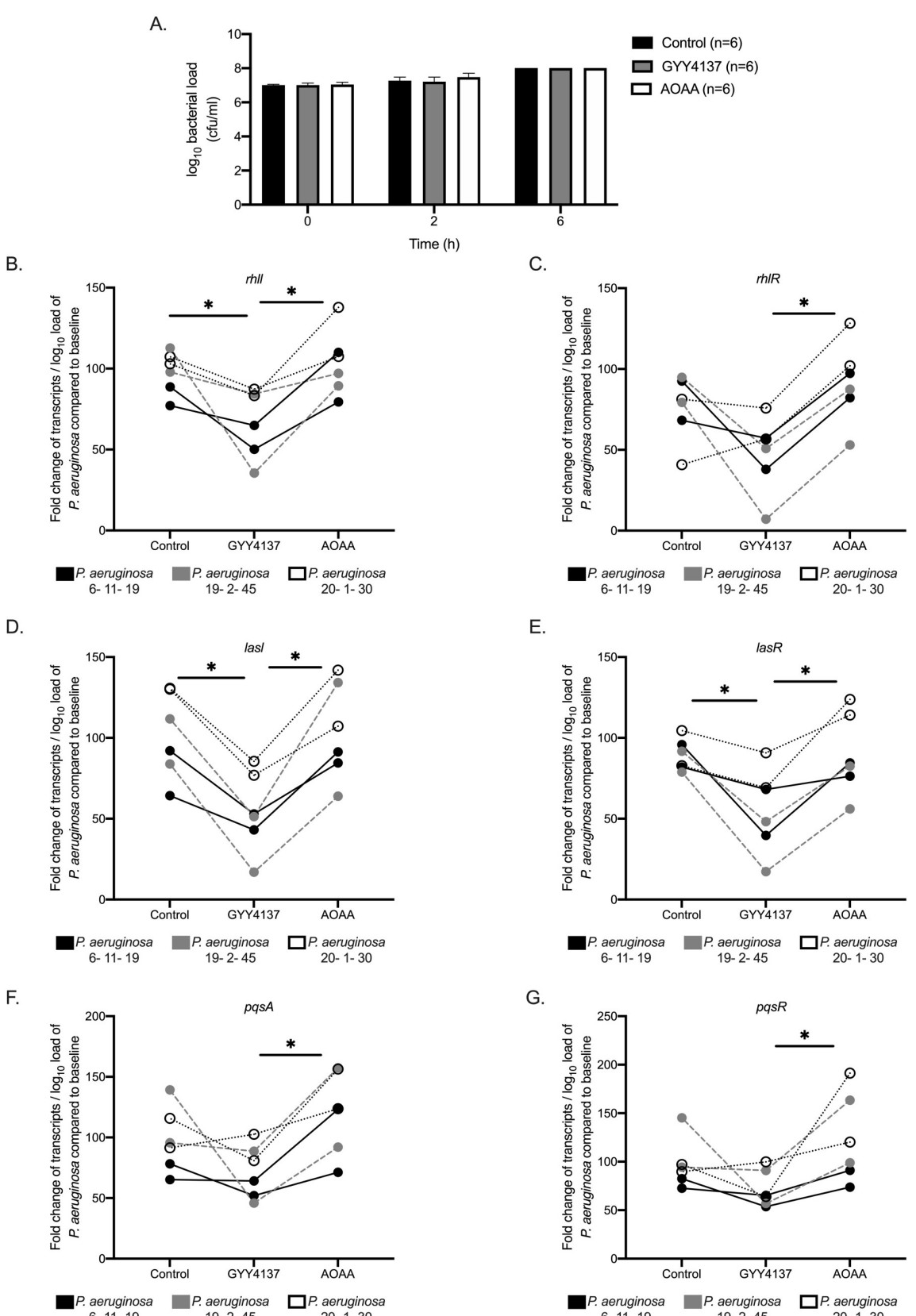

**Fig 5. Modulation of growth and expression of the quorum sensing (QS) genes of multi-drug resistant (MDR) *Pseudomonas aeruginosa* in vitro.** MDR *P. aeruginosa* isolates 6–11–19; 19–2–45 and 20–1–30 were incubated for 2 and 6 hours with/ without $H_2S$ donor GYY4137 and $H_2S$ biosynthesis inhibitor aminooxyacetic acid (AOAA). Experiment was performed in duplicate. A) Bacterial growth at 0, 2 and 6 hours on incubation; B-G) Fold change of number of transcripts of QS genes *rhll, rhlR, lasl, lasR, pqsA, pqsR* per $\log_{10}$ of MDR *P. aeruginosa* for each isolate is shown. Comparison by the ANOVA test with Bonferroni correction for multiple comparisons; Only statistically significant values are shown; * $p < 0.05$.

cell- to cell communication and evasion of the host innate immune response. Additionally, a negative correlation was found between lung $H_2S$ levels and the expression of QS gene *rhll, rhlR, lasl, lasR* and *pqsA* in *Cse* $^{+/+}$ mice but not in *Cse* $^{-/-}$ mice. Subsequently, $H_2S$ produced specifically through CSE acts as a restrain of QS activation, rendering MDR *P. aeruginosa* susceptible to phagocytosis.

Previous studies show that garlic extract, which contains several polysulfides, which in vivo stimulate the synthesis of hydrogen sulfide, inhibits the quorum sensing system of *P. aeruginosa* rendering it vulnerable to elimination through polymorphonuclear cells [42–44]. However, the exact mechanism by which $H_2S$ regulates the expression of the quorum sensing genes of *P. aeruginosa* still remains unclear. AHLs have been shown to reduce the activation of the nuclear factor-kappa β (NF-κB) pathway, which in term leads to a deficient innate immune response [45,46]. On the contrary, $H_2S$ produced by CSE has been found to up- regulate the expression and actions of NF-κB, through sulfhydration [47]. Thus, the up-regulation of the NF-κB pathway mediated by $H_2S$ is a potential mechanism for the limitation of the activation of the quorum sensing system of *P. aeruginosa*.

The importance of $H_2S$ is even more complex, when one considers that bacteria are themselves capable of producing $H_2S$, through the same enzymatic systems as eukaryotic organisms. *In vitro* data show that *Escherichia coli* can use the $H_2S$ they produce, as an autoinducer of bacterial growth *[48]*. It seems that $H_2S$ is also necessary for the maintenance of bacterial resistance against antimicrobial treatment. *Staphyloccocus aureus*, *Escherichia coli*, *Bacillus anthracis* and *P.aeruginosa* strains, in which the genes that express the $H_2S$ producing enzymes *Cse*, *Cbs* and *3Mst* were genetically silenced, were rendered susceptible to previously ineffective antibiotics *in vitro* [49,50] as well as to elimination by the host immune system in spleen co-culture experiments *in vitro* and in mouse models of wound infection *in vivo* [27]. Yet other $H_2S$-related mechanisms apply to the elimination of *Mycobacteria* by the immune system: *M. tuberculosis* appears to coopt the host $H_2S$ production and use it to drive its own metabolism and proliferation: in *M. tuberculosis* infected mice, while also upregulating host macrophage CSE expression to a level where the $H_2S$ produced begins to suppress the immune function of the macrophage: under these conditions, deletion of host CBS or CSE suppresses bacterial growth and improves the survival of the infected mice *[51,52]*. The above examples illustrate that the role of host and bacterial $H_2S$ in infection is pathogen and context-dependent, and this must be kept in mind when considering future translational efforts centered on the pharmacological modulation of $H_2S$ homeostasis.

The translation into the clinical setting of findings of pre-clinical studies on the role of $H_2S$ in infection poses a great challenge, mainly because of the great variance of measured $H_2S$ levels in the plasma, which ranges from 2μM to over 500μM [6,53–55]. The measured levels seem to be disease- and pathogen- specific and also depend on the measuring method. This variance renders the use of a general threshold of $H_2S$ plasma levels difficult and necessitates a more personalized approach.

In conclusion, we can show that *Cse*-derived host $H_2S$ is a defense mechanism that acts via reduction of host pathogen load, affording resistance to infection. $H_2S$ is also a possible biomarker for the severity and outcome of MDR *P. aeruginosa* infections and could be utilized as

A.

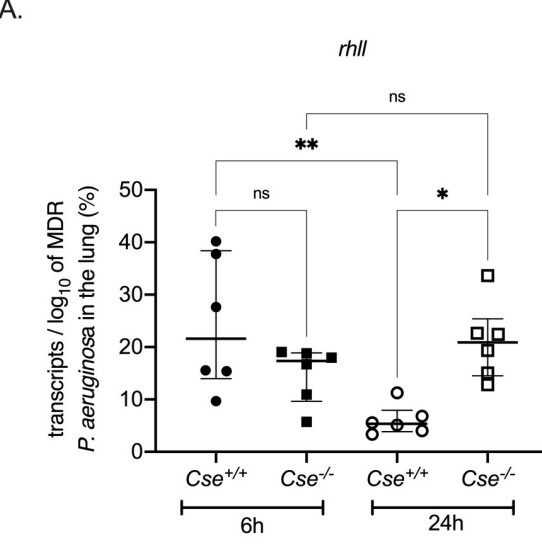

B.

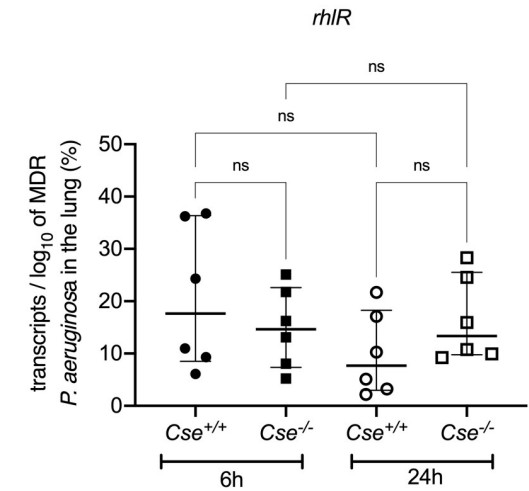

C.

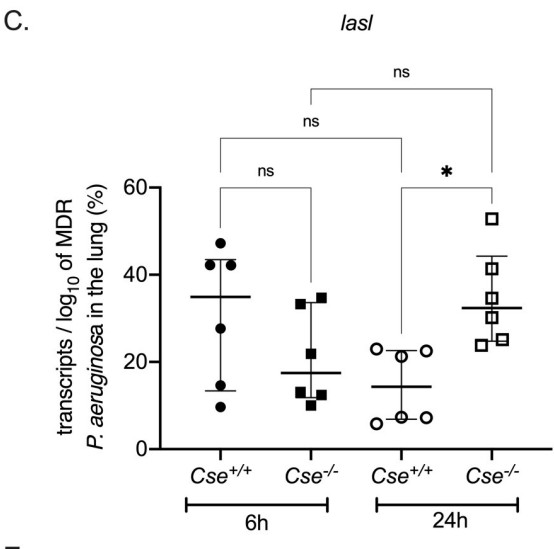

D.

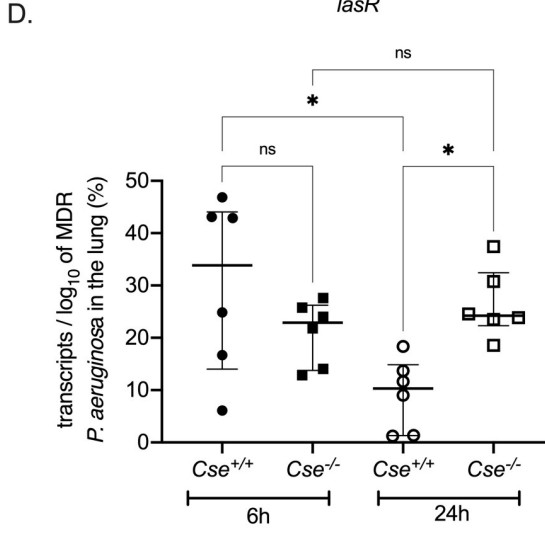

E.

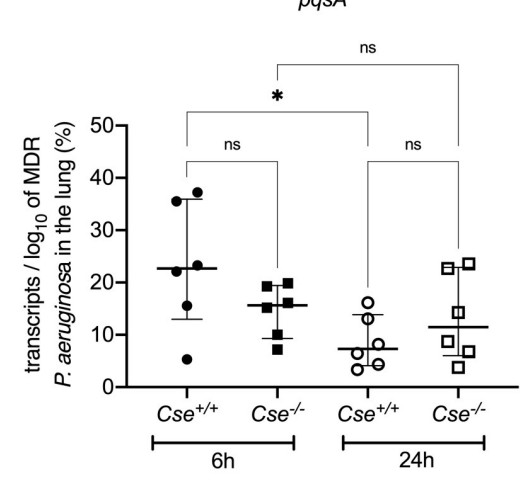

F.

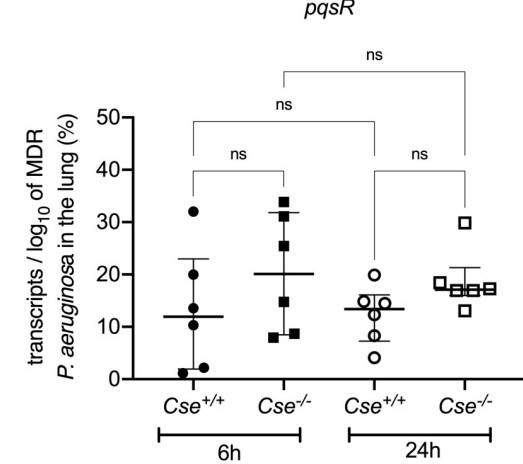

**Fig 6. Importance of endogenous H$_2$S in the expression of quorum sensing (QS) genes of multi-drug resistant (MDR)** *Pseudomonas aeruginosa* **in vivo.** A-F) Transcripts of QS genes *rhlI, rhlR, lasI, lasR, pqsA, pqsR* per log$_{10}$ of MDR *P. aeruginosa* in the lung in *Cse*$^{+/+}$ and *Cse*$^{-/-}$ mice, sacrificed 6 and 24h after experimental infection (6 mice per group/ timepoint). Comparisons by the ANOVA test with Bonferroni correction for multiple comparisons; ns non-significant, $^*$ p$<$ 0.05.

an adjunctive criterion, together with other established biomarkers like the SOFA score, CRP and PCT for decision making. The mechanism resides on the inhibition of the expression of the QS system of *P. aeruginosa*, rendering it susceptible to an efficient neutrophil guided phagocytosis.

## Supporting information

**S1 Fig. Patient selection process.** Description of the selection steps for the three groups of comparison. Abbreviations: VAP: ventilator associated pneumonia; TBS: tracheobronchial secretions.
(TIF)

**S2 Fig. Serum H$_2$S decreases in bacterial ventilator associated pneumonia.** Serum levels of H$_2$S in healthy volunteers and on day 1 among patients with sepsis due to VAP were measured by HPLC. Comparisons by the Mann Whitney U test); $^{**}$ p$<$ 0.01.
(TIF)

**S3 Fig. Effect of H$_2$S production deficiency on the outcome of infection by** *P. aeruginosa*. A) DNA genomic patterns of *P. aeruginosa* isolates 6–11–19 (P1); 19–2–45 (P2) and 20–1–30 (P3) determined by pulse field gel electrophoresis. L: DNA Ladder. Survival analysis between B) *Cse*$^{+/+}$ and *Cse*$^{-/-}$ mice after infection with MDR *P. aeruginosa* isolate 19–2–45; C) *Cse*$^{+/+}$ and *Cse*$^{-/-}$ mice after infection with MDR *P. aeruginosa* isolate 20–1–30; Results of the log-rank test and the relevant p- values are given. D) Bacterial load (cfu/g) in the liver and in the lung of *Cse*$^{+/+}$ and *Cse*$^{-/-}$ mice after infection with MDR *P. aeruginosa* isolate 19–2–45; E) Bacterial load (cfu/g) in the liver and in the lung of *Cse*$^{+/+}$ and *Cse*$^{-/-}$ mice after infection with MDR *P. aeruginosa* isolate 20–1–30. Comparisons by the Mann Whitney U test. $^*$ p$<$ 0.05, $^{**}$ p$<$ 0.01.
(TIF)

**S4 Fig. Lack of effect of host H$_2$S on production of tumor necrosis factor alpha (TNFα) after multi- drug resistant (MDR)** *Pseudomonas aeruginosa* **infection.** *Cse*$^{+/+}$ and *Cse*$^{-/-}$ mice (n = 6 per group per timepoint) were sacrificed 6 and 24 hours after experimental infection by MDR *P. aeruginosa* isolate 6–11–19. Concentration of TNFα A) in serum, B) tissue supernatants and C) supernatants of stimulated splenocytes (stimuli medium, LPS from *Escherichia coli* O55:B5 for 24h and *C. albicans* for 5 days). Comparison by the Mann Whitney U test; ns non- significant.
(TIF)

**S5 Fig. Lack of effect of endogenous H$_2$S on the production of pro- and anti- inflammatory cytokines after multi- drug resistant (MDR)** *Pseudomonas aeruginosa* **infection.** *Cse*$^{+/+}$ and *Cse*$^{-/-}$ mice (n = 6) were sacrificed 6 hours after experimental infection by MDR *P. aeruginosa* isolate 6–11–19. Concentration of IL-1α, IL-1β, IL-6, IL-10, IL-12, IL-27, IFNγ, IFNβ in serum. Comparison by the Mann Whitney U test; ns non- significant.
(TIF)

**S6 Fig. Absence of correlation of the CSE enzyme with baseline blood cell counts.** Healthy *Cse*$^{+/+}$ and *Cse*$^{-/-}$ mice (n = 8 per group) were sacrificed. Absolute counts of A) red blood cells (RBCs); B) platelets; C) white blood cells (WBCs); D) neutrophils and E) lymphocytes were

determined. Comparison by the Mann Whitney U test; ns non- significant.
(TIF)

**S7 Fig. Absence of effect of bone marrow transplantation on serum H₂S level.** H$_2$S levels in serum of naïve $Cse^{+/+}$ and $Cse^{-/-}$ mice before and after bone marrow transplantation (BMT) were measured bu high- performance liquid chromatography (HPLC). Comparison by the ANOVA test with Bonferroni correction for multiple comparisons; ns non- significant, $^*$ p$<$ 0.05, $^{**}$ p$<$ 0.01.
(TIF)

**S8 Fig. Neutrophil activity against multi-drug resistant (MDR) *Pseudomonas aeruginosa* is mediated by the CSE enzyme.** $Cse^{+/+}$ and $Cse^{-/-}$ mice (n = 6 per group per timepoint) were sacrificed 6, 12 and 24 hours after experimental infection by MDR *P. aeruginosa* isolate 6–11–19. Correlations between bacterial outgrowth and MPO in the liver and in the lung for each group. Spearmann rank correlation coefficient ($r_s$), relevant p- value and interpolation line for each group are given.
(TIF)

**S9 Fig. Modulation through CSE derived H₂S of the quorum sensing (QS) system of multi-drug resistant (MDR) *Pseudomonas aeruginosa*.** A-F) Correlation between Transcripts of QS genes *rhII, rhIR, lasI, lasR, pqsA, pqsR* per log$_{10}$ of MDR *P. aeruginosa* in the lung and H$_2$S levels in the lung in $Cse^{+/+}$ and $Cse^{-/-}$ mice, sacrificed 6 after experimental infection by MDR *P. aeruginosa* isolate 6–11–19. (6 mice per group). Spearmann rank correlation coefficient ($r_s$), interpolation line for each group and relevant p- value are given.
(TIF)

**S1 Text. Table A in S1 Text.** Baseline and clinical characteristics of patients with sepsis due to VAP grouped by pathogen Table B in S1 Text: Mobile phase gradient of HPLC Table C in S1 Text: Quorum sensing (QS) genes and primers of *P. aeruginosa* isolates used for qRT-PCR.
(DOC)

## Author Contributions

**Conceptualization:** Andreas Papapetropoulos, Evangelos J. Giamarellos-Bourboulis.

**Data curation:** Georgios Renieris, Dionysia-Eirini Droggiti, Konstantina Katrini, Panagiotis Koufargyris, Theologia Gkavogianni, Eleni Karakike, Nikolaos Antonakos, Georgia Damoraki, Athanasios Karageorgos, Labros Sabracos, Antonia Katsouda, Elisa Jentho, Sebastian Weis, Rui Wang, Michael Bauer, Csaba Szabo, Kalliopi Platoni, Vasilios Kouloulias, Andreas Papapetropoulos.

**Formal analysis:** Georgios Renieris, Dionysia-Eirini Droggiti, Konstantina Katrini, Panagiotis Koufargyris, Theologia Gkavogianni, Eleni Karakike, Nikolaos Antonakos, Georgia Damoraki, Athanasios Karageorgos, Labros Sabracos, Antonia Katsouda, Elisa Jentho, Sebastian Weis, Rui Wang, Michael Bauer, Csaba Szabo.

**Funding acquisition:** Evangelos J. Giamarellos-Bourboulis.

**Investigation:** Georgios Renieris, Konstantina Katrini, Panagiotis Koufargyris, Theologia Gkavogianni, Eleni Karakike, Nikolaos Antonakos, Elisa Jentho, Sebastian Weis, Michael Bauer, Kalliopi Platoni.

**Methodology:** Georgios Renieris, Dionysia-Eirini Droggiti, Eleni Karakike, Labros Sabracos, Sebastian Weis, Rui Wang, Michael Bauer, Csaba Szabo, Vasilios Kouloulias, Evangelos J. Giamarellos-Bourboulis.

**Project administration:** Andreas Papapetropoulos, Evangelos J. Giamarellos-Bourboulis.

**Resources:** Evangelos J. Giamarellos-Bourboulis.

**Software:** Georgios Renieris.

**Supervision:** Rui Wang, Csaba Szabo, Andreas Papapetropoulos, Evangelos J. Giamarellos-Bourboulis.

**Validation:** Sebastian Weis, Rui Wang, Csaba Szabo, Andreas Papapetropoulos, Evangelos J. Giamarellos-Bourboulis.

**Visualization:** Georgios Renieris, Labros Sabracos, Michael Bauer, Evangelos J. Giamarellos-Bourboulis.

**Writing – original draft:** Georgios Renieris.

**Writing – review & editing:** Dionysia-Eirini Droggiti, Konstantina Katrini, Panagiotis Koufargyris, Theologia Gkavogianni, Eleni Karakike, Nikolaos Antonakos, Georgia Damoraki, Athanasios Karageorgos, Labros Sabracos, Antonia Katsouda, Elisa Jentho, Sebastian Weis, Rui Wang, Michael Bauer, Csaba Szabo, Kalliopi Platoni, Vasilios Kouloulias, Andreas Papapetropoulos, Evangelos J. Giamarellos-Bourboulis.

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
