## [Decision Letter · Decision Letter 0]

22 Dec 2020

Dear Prof. Giamarellos-Bourboulis,

Thank you very much for submitting your manuscript "HOST CYSTATHIONINE- γ  LYASE DERIVED HYDROGEN SULFIDE PROTECTS AGAINST PSEUDOMONAS AERUGINOSA SEPSIS" for consideration at PLOS Pathogens. As with all papers reviewed by the journal, your manuscript was reviewed by members of the editorial board and by several independent reviewers. In light of the reviews (below this email), we would like to invite the resubmission of a significantly-revised version that takes into account the reviewers' comments. Please pay particular attention to Reviewer 2's concerns and address each with either additional experiments/revisions or an explanation for why you feel such additions are not necessary.

We cannot make any decision about publication until we have seen the revised manuscript and your response to the reviewers' comments. Your revised manuscript is also likely to be sent to reviewers for further evaluation.

Sincerely,

Matthew C Wolfgang

Associate Editor

PLOS Pathogens

Alan Hauser

Section Editor

PLOS Pathogens

Kasturi Haldar

Editor-in-Chief

PLOS Pathogens

orcid.org/0000-0001-5065-158X

Michael Malim

Editor-in-Chief

PLOS Pathogens

orcid.org/0000-0002-7699-2064

Reviewer's Responses to Questions

**Part I - Summary**

Reviewer #1: This is an interesting study investigating the role of H2S in sepsis. This pathway has not been previously investigated in these clinical models of disease and in patients with sepsis, and the results are novel and interesting. The combination between experimental and clinical data is particularly compelling, arguing for the importance of the findings.

Reviewer #2: The manuscript by G. Renieris investigate the role of H2S production by CSE ininfections by Pseudomonas aeruginosa. First the authors show that in human serum H2S is lower in severe case of sepsis du to VAP by. P. aeruginosa. This is somehow specific to this pathogen as the benefits of H2S were not observed in sepsis caused by other pathogens. Importantly, in mice, data show that mortality is higher in mice lacking CSE (whole body KO), and could be restored by exogenous administration of H2S via STS or GYY. It is a welcome choice to use STS as this drug is clinically approved, although its role as an H2S donor still need to be clearly demonstrated. Next, the authors investigate the mechanisms that lead to death in the case of host H2S deficiency. First, they show that in mice rendered neutropenic by cyclophosphamide, the survival benefit of CSE was lost. Next, they observed increase bacterial load in CSE KO mice, attributed to decrease phagocytic capacity. Finally, the authors showed that the specific susceptibility of CSE KO mice to P.aeruginosa infection was due to an H2S dependent down-regulation of quorum sensing genes of P.aeruginosa.

Taken together, this is an interesting study that addressed the role H2S in P. aeruginosa and sepsis. This represent novel findings, with potentially important clinical implication (as stated by the authors “Pseudomonas (P.) aeruginosa are responsible for more than 50% of VAP cases in Greece and are characterized by high rates of antimicrobial resistance”). While some of the data are very convincing, there are some areas that require additional experiments to address the key questions and to potentially support the conclusions. This would also be needed before it is suited for publication in this journal. As the neutrophil seem to be critical cellular mediator of the phenotype in CSE KO mice, and since CSE is a promiscuous enzyme, bonne marrow transfer (or CSE removal in neutrophil specifically) would be valuable. I am also wondering what the impact on serum H2S in this setting would be. Next, and from a clinical perspective it would be interesting to see the effect of H2S administration in WT mice (survival after P.aeruginosa exposure). In line with this, I would be curious to see if H2S administration after the infection could rescue the mice. Finally, it remains unclear by which mechanism H2S targets quorum sensing genes of P.aeruginosa (could it be linked to linked sulfhydration of NF-κB, PMID: 22244329).

**Part II – Major Issues: Key Experiments Required for Acceptance**

Reviewer #1: Major comments

1. The quality of the figures is poor, that should be improved.

2. The bacterial load in the CSE+/+ and CSE-/- mice after Pseudomonas infection is given in Fig.4F at the spleen level. Were bacterial burdens assessed also in other organs more likely to be important for determining mortality?

Reviewer #2: As the neutrophil seem to be critical cellular mediator of the phenotype in CSE KO mice, and since CSE is a promiscuous enzyme, bonne marrow transfer (or CSE removal in neutrophil specifically) would be valuable. I am also wondering what the impact on serum H2S in this setting would be. Next, and from a clinical perspective it would be interesting to see the effect of H2S administration in WT mice (survival after P.aeruginosa exposure). In line with this, I would be curious to see if H2S administration after the infection could rescue the mice. Finally, it remains unclear by which mechanism H2S targets quorum sensing genes of P.aeruginosa (could it be linked to linked sulfhydration of NF-κB, PMID: 22244329). The following points/questions should also be address in more detail:

1. Figure 1 and methods: precise where and how the heathy serum was obtained. What are their baseline comorbidities (if any)? Are they matched for anything (e.g. age )?

2. Figure 1A. It appears that the H2S levels in the survivors’ group from P. aeruginosa infection are significantly higher that the others. Can you provide some reasonable explanation?

3. Could you pool the data from the sepsis group, which would indicate lower H2S levels in patients with sepsis compare to healthy patients? This is somehow counterintuitive as H2S seems to increase upon infection in mice (Figure 3)

4. Figure S1. The Yes / No box significance isn’t clear.

5. Figure 1B. Could you provide the H2S levels value distribution for all of the sample? How do you explain that H2S values between studies, using the same methods (PMID: 32433216) differs greatly (x100 fold difference)? I believe that threshold would therefore be difficult to apply in the clinical setting. I also believe that H2S could be sensitive at predicting survival, but it is hard to believe that it would be specific since it has pleiotropic role. Also other score might be more specific (e.g. SOFA).

6. Add the # at risk to every Kaplan Meier data.

7. How do you explain that survival is different in CSE KO mice from Fig 2A and B. Again provide the number at risk. Can you explain why there is some mice that seems to be long term survivors?

8. The model used here seems to be very severe as 50% of WT mice are dead at 24hrs. This is largely above what’s observed in human. This is even more concerning in Figure 3F using the LPS model. Can it be that CSE provides some “unspecific survival benefits”. The importance of CSE in longevity and stress resistance was demonstrated in several studies (e.g. Hine et al, Cell 2015).

9. Fig 3. CSE KO have significant amount of baseline H2S levels. Also H2S seems to increase upon infection also in the KO too. Is this increase significant ? I would suggest to change the X axis, and group the data by genotype (and do so in all of the graph).

10. Figure 3D. Please provide serum H2S levels in 3MST WT and KO.

11. Please provide cytokine value at baseline and not only after infection. This is of particular interest in CSE WT and KO.

**Part III – Minor Issues: Editorial and Data Presentation Modifications**

Reviewer #1: Minor suggestions

1. There is probably a mistake in the sentence: “The ratio of CD11b+CD11c-CD45 to CD11b+CD11c-CD45 of the spleen, which mostly represent the ratio of the granulocyte to the macrophage population [28]”, as I guess different cell markers need to be used for the two populations.

2. The authors conclude based on the circulating cytokine measurements that “the protective role is not associated with an effect on cytokine production by monocytes or lymphocytes”. However, one paragraph later they show decreased production of mediators produced by neutrophils. It is indeed cytokine production locally which is more important, rather than circulating cytokine concentration in the blood, so I would suggest to change this conclusion.

3. Can the authors speculate about the functionality of the H2S system in patients with cystic fibrosis?

Reviewer #2: - The abstract should be revised to be more easily read and clearer.

- Spell H2S and not H2S.

- Figure 4E legends: Ratio of CD11b+CD11cCD45 cells to CD11b+CD11cCD45 cells doesn’t make sense.

PLOS authors have the option to publish the peer review history of their article (what does this mean?). If published, this will include your full peer review and any attached files.

Reviewer #1: No

Reviewer #2: No
---

## [Decision Letter · Decision Letter 1]

12 Mar 2021

Dear Prof. Giamarellos-Bourboulis,

We are pleased to inform you that your manuscript 'HOST CYSTATHIONINE-γ LYASE DERIVED HYDROGEN SULFIDE PROTECTS AGAINST PSEUDOMONAS AERUGINOSA SEPSIS' has been provisionally accepted for publication in PLOS Pathogens.

Best regards,

Matthew C Wolfgang

Associate Editor

PLOS Pathogens

Alan Hauser

Section Editor

PLOS Pathogens

Kasturi Haldar

Editor-in-Chief

PLOS Pathogens

orcid.org/0000-0001-5065-158X

Michael Malim

Editor-in-Chief

PLOS Pathogens

orcid.org/0000-0002-7699-2064

Reviewer Comments (if any, and for reference):

Reviewer's Responses to Questions

**Part I - Summary**

Reviewer #2: The authors added significant amount of work, which address all of my previous concerns.

**Part II – Major Issues: Key Experiments Required for Acceptance**

Reviewer #2: none

**Part III – Minor Issues: Editorial and Data Presentation Modifications**

Reviewer #2: none

PLOS authors have the option to publish the peer review history of their article (what does this mean?). If published, this will include your full peer review and any attached files.

Reviewer #2: No

---

## [Editor Report · Acceptance letter]

23 Mar 2021

Dear Prof. Giamarellos-Bourboulis,

We are delighted to inform you that your manuscript, "HOST CYSTATHIONINE-γ LYASE DERIVED HYDROGEN SULFIDE PROTECTS AGAINST PSEUDOMONAS AERUGINOSA SEPSIS," has been formally accepted for publication in PLOS Pathogens.

Best regards,

Kasturi Haldar

Editor-in-Chief

PLOS Pathogens

orcid.org/0000-0001-5065-158X

Michael Malim

Editor-in-Chief

PLOS Pathogens

orcid.org/0000-0002-7699-2064